# EvoMesh: Adaptive Physical Simulation with Hierarchical Graph Evolutions

**Huayu Deng** [1]  **Xiangming Zhu** [1]  **Yunbo Wang** [1]  **Xiaokang Yang** [1]

## Abstract

Graph neural networks have been a powerful tool for mesh-based physical simulation. To efficiently model large-scale systems, existing methods mainly employ hierarchical graph structures to capture multi-scale node relations. However, these graph hierarchies are typically manually designed and fixed, limiting their ability to adapt to the evolving dynamics of complex physical systems. We propose EvoMesh, a fully differentiable framework that jointly learns graph hierarchies and physical dynamics, adaptively guided by physical inputs. EvoMesh introduces *anisotropic message passing*, which enables direction-specific aggregation of dynamic features between nodes within each hierarchy, while simultaneously learning node selection probabilities for the next hierarchical level based on physical context. This design creates more flexible message shortcuts and enhances the model's capacity to capture long-range dependencies. Extensive experiments on five benchmark physical simulation datasets show that EvoMesh outperforms recent fixed-hierarchy message passing networks by large margins. The project page is available at https://hbell99.github.io/evo-mesh/.

## 1. Introduction

Simulating physical systems with deep neural networks has achieved remarkable success due to their efficiency compared with traditional numerical solvers. Graph Neural Networks (GNNs) have been validated as a powerful tool for mesh-based simulation, such as for fluids and rigid collisions (Wu et al., 2020). The primary mechanism driving the GNN-based models is message passing, where time-varying physical quantities are encoded within the mesh structure and are temporally updated by aggregating information

*Table 1.* **Comparison of mesh-based physical simulation models.** *Dynamic hierarchy* refers to hierarchical graph structures that evolve over time. *Adaptive* indicates that the graph structures are determined by physical inputs. *Prop.* denotes feature propagation.

| Model | Dynamic Hierarchy | Adaptive Hierarchy | Anisotropic Intra-level Prop | Learnable Inter-level Prop |
|---|:---:|:---:|:---:|:---:|
| MGN (2021) | ✗ | ✗ | ✗ | ✗ |
| Lino *et al.* (2022) | ✗ | ✗ | ✗ | ✓ |
| BSMS (2023) | ✗ | ✗ | ✗ | ✗ |
| Eagle (2023) | ✗ | ✗ | ✓ | ✓ |
| HCMT (2024) | ✗ | ✗ | ✓ | ✗ |
| EvoMesh | ✓ | ✓ | ✓ | ✓ |

broadcast from neighboring nodes (Sanchez-Gonzalez et al., 2020; Pfaff et al., 2021; Allen et al., 2023). Existing methods generally rely on repeated local message passing to propagate influence over long distances, which becomes extremely costly for large-scale mesh graphs. A common solution involves using multi-scale graph structures to create direct information shortcuts between distant nodes. (Lino et al., 2022; Cao et al., 2023; Yu et al., 2024; Han et al., 2022; Fortunato et al., 2022).

However, as shown in Table 1, previous methods commonly rely on heuristic node selection to create predefined (data-independent) coarser message passing graphs (Cao et al., 2023; Yu et al., 2024). These predefined graphs limit the model's adaptation ability in two key ways. First, the fixed graph hierarchies, applied to the entire input sequence, do not account for the variety of physical contexts. In practical systems like turbulence, even with identical boundary conditions, small changes in initial conditions can lead to significant differences in subsequent dynamics. Second, since the spatial correlations in a physical process can evolve over time, static graph hierarchies are insufficient for capturing the time-varying node interactions.

To tackle this challenge, we propose a novel neural network approach named EvoMesh, which constructs data-adaptive and time-evolving graph hierarchies based on the input physical quantities. The key insight is to develop a differentiable node selection method that allows for flexible correlation of long-range, dynamic node interactions. This is technically supported by an *anisotropic message passing* (AMP) mechanism, which (i) aggregates neighboring features with non-uniform, learnable importance weights within each hi-

---

[1]MoE Key Lab of Artificial Intelligence, AI Institute, Shanghai Jiao Tong University. Correspondence to: Yunbo Wang <yunbow@sjtu.edu.cn>.

*Proceedings of the 42nd International Conference on Machine Learning*, Vancouver, Canada. PMLR 267, 2025. Copyright 2025 by the author(s).

erarchical level, (ii) predicts the probabilities of the node being retained for the next hierarchy based on physical context, and (iii) adaptively learns cross-hierarchy interactions to optimize information flow across scales. To enable differentiability in the node selection process, we approximate the discrete downsampling decisions using Gumbel-Softmax.

Another advantage of the AMP mechanism is its ability to enable features to transfer between nodes with varying importance, aligning with the directionally non-uniform nature of the dynamic patterns, as observed in scenarios such as CylinderFlow, AirFoil, and Flying Flag simulations. It applies to both *intra-level* and *inter-level* feature propagation. In contrast, as shown in Table 1, most previous GNN-based mesh simulation methods perform *isotropic* feature aggregation within the intra-level transition and rely on unlearnable importance weights to transfer inter-level information across hierarchical levels, assuming equal contributions from neighboring nodes.

Overall, our contributions are summarized as follows:

- We present EvoMesh, which generates dynamic graph hierarchies through differentiable node selection, enabling adaptive modeling of multi-scale physical relations.

- EvoMesh employs anisotropic message passing to enable directionally varied feature propagation both within and across graph hierarchies.

- On average, EvoMesh outperforms fixed-hierarchy models by around $20\%$ across a range of standard benchmarks. It also demonstrates strong generalization to test cases with time-varying mesh structures, novel resolutions, and out-of-distribution dynamics.

## 2. Preliminaries

**Message passing.** We consider simulating mesh-based physical systems, where the task is to predict the dynamic quantities of the mesh at future timesteps given the current mesh configuration. A mesh-based system is represented as a bi-directed graph $\mathcal{G} = (\mathcal{V}, \mathcal{E})^1$, where $\mathcal{V}$ and $\mathcal{E}$ denote the set of nodes and edges, respectively. *Message passing neural networks* (MPNNs) compute the node representations by stacking multiple message passing layers of the form:

$$\text{Edge update: } \hat{\mathbf{e}}_{ij} = \phi^e(\mathbf{e}_{ij}, \mathbf{v}_i, \mathbf{v}_j); \qquad (1)$$

$$\text{Node update: } \hat{\mathbf{v}}_i = \phi^v\left(\mathbf{v}_i, \psi\left(\{\hat{\mathbf{e}}_{ij} \mid \forall j, e_{ij} \in \mathcal{E}\}\right)\right), \quad (2)$$

where $\mathbf{v}_i$ is the feature of node $v_i \in \mathcal{V}$ and $\psi$ denotes a non-parmatric aggregation function. The function $\phi^e$ updates the features of edges based on the endpoints, while $\phi^v$

---

¹*Bi-directed* means each original undirected edge is represented twice in $\mathcal{G}$: if there is an edge between $i$ and $j$, it is represented as two directed edges $i \rightarrow j$ and $j \rightarrow i$. Each node has a self-loop.

updates the node states with aggregated messages from its neighbors. In existing GNN-based mesh simulation methods, multi-layer perceptrons (MLPs) with residual connections are commonly employed for $\phi^e(\cdot)$ and $\phi^v(\cdot)$, with the ***non-parametric*** aggregation function $\psi(\cdot)$ being defined as the sum of edge features. Notably, since the aggregation function treats all neighbors equally, the contributions from neighboring nodes may be averaged out, and the repeated message-passing process can further dilute distinctive node features. This issue is exacerbated in dynamic physical systems, where transferring directed patterns is crucial. Attention-based methods address this issue by reweighting neighbor features, either locally or globally (Veličković et al., 2018; Yu et al., 2024; Han et al., 2022; Yun et al., 2019). While effective for directional aggregation, most weighting remains limited to intra-level features and does not support dynamic graph hierarchy construction.

**Hierarchical MPNNs.** To facilitate long-range modeling, hierarchical MPNNs process information at $L$ scales by creating a graph for each level and propagating information between them (Lino et al., 2022; Fortunato et al., 2022; Cao et al., 2023; Yu et al., 2024). Let $\mathcal{G}_1 = (\mathcal{V}_1, \mathcal{E}_1)$ represent the graph structure at the finest level, *i.e.*, the input mesh. The lower-resolution graphs $\mathcal{G}_2, \mathcal{G}_3, \ldots, \mathcal{G}_L$, with $|\mathcal{V}_1| > |\mathcal{V}_2| > \ldots > |\mathcal{V}_L|$, contain fewer nodes and edges, which allows for more efficient feature propagation over longer physical distances with certain propagation steps. The typical process for constructing multi-scale structures primarily involves downsampling and upsampling between adjacent graph hierarchies. Downsampling reduces the number of nodes while upsampling transfers information from a lower-resolution graph to a higher-resolution one. The downsampling operation includes two steps:

- SELECT: Nodes are selected from the current graph structure $\mathcal{G}_l$ to create a new, coarser graph $\mathcal{G}_{l+1}$. Various strategies have been proposed to construct $\mathcal{V}_{l+1}$, including hand-crafted designs (Lino et al., 2022; Cao et al., 2023; Yu et al., 2024), geometric clustering (Han et al., 2022; Janny et al., 2023), and differentiable pooling methods that predict cluster assignments or select top-ranked informative nodes (Ying et al., 2018; Gao & Ji, 2019; Lee et al., 2019; Ranjan et al., 2020). The edges $\mathcal{E}_{l+1}$ in $\mathcal{G}_{l+1}$ are constructed by connecting the selected nodes based on the original edges $\mathcal{E}_l$. However, this process can sometimes lead to loss of connectivity and introduce partitions (Gao & Ji, 2019; Lee et al., 2019; Cao et al., 2023). To mitigate this, connectivity in $\mathcal{E}_{l+1}$ can be strengthened by adding $K$-hop edges.

- REDUCE: The features of the nodes in $\mathcal{V}_{l+1}$ are aggregated from their corresponding neighborhood features in the finer graph $\mathcal{G}_l$.

The upsampling process is represented by EXPAND, which

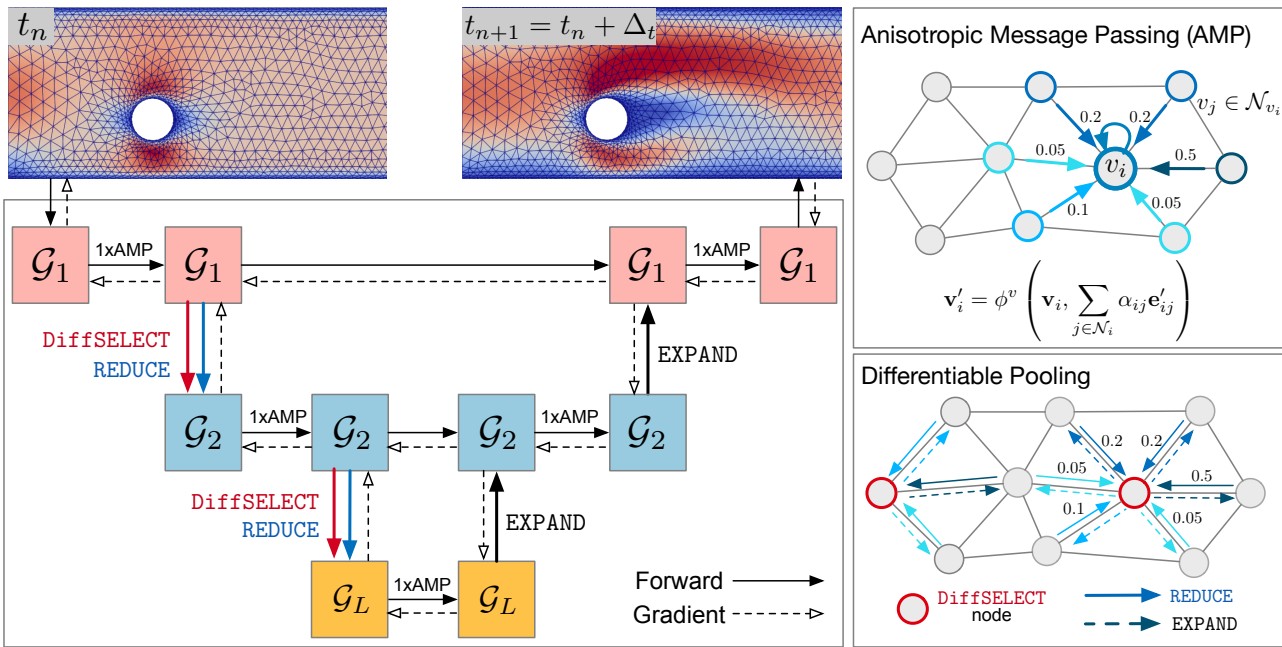

Figure 1. **The architecture of EvoMesh.** Physical dynamics is modeled on multiple graph resolutions with adaptive structures, $\mathcal{G}_1, \mathcal{G}_2, \ldots, \mathcal{G}_L$, and are processed using their respective AMP layers. The `DiffSELECT` operation performs differentiable pooling to create coarser graphs with learnable downsampling probabilities. `REDUCE` and `EXPAND` integrate inter-level information using learned feature aggregation weights over the neighboring nodes. EvoMesh is trained end-to-end with one-step supervision.

is the inverse of the `REDUCE` function and transfers information from the coarser level back to the finer level. Most previous work generates coarser graphs either by using numerical software or by downsampling the input mesh through heuristic pooling strategies (Cao et al., 2023; Lino et al., 2022; Yu et al., 2024; Janny et al., 2023). This process is performed during the data preprocessing stage. The preprocessed hierarchy with the same input mesh topology is reused across different initial conditions and time steps.

## 3. Method

In this section, we introduce EvoMesh, a fully differentiable model that adaptively generates time-evolving graph hierarchies over the sequence, while simultaneously simulating the physical system over these learned hierarchical graphs. Figure 1 demonstrates an overview of the proposed model, which operates in an *encode-process-decode* pipeline. The encoder first maps the input field to a latent feature space $\mathbf{V}_1 = \{\mathbf{v}_i | v_i \in \mathcal{V}_1\}$ at the original mesh resolution. Subsequently, we model the physical dynamics across the learned multi-scale graph hierarchies with adaptive graph structures.

In Section 3.1, we present the details of the AMP layer. In Section 3.2, we discuss the approach for learning context-aware graph hierarchies. In Section 3.3, we describe the inter-level downsampling and upsampling processes that incorporate AMP-based feature propagation. Finally, in

Section 3.4, we outline the implementation details.

### 3.1. Anisotropic Message Passing

We introduce the AMP layer, which facilitates information propagation both within and between graph hierarchies, enabling EvoMesh to effectively capture local and long-range dependencies simultaneously.

As shown in Eq. (2), a common non-parametric aggregation in GNN-based mesh simulation is to use the summation for node update: $\hat{\mathbf{v}}_i = \phi^v\Big(\mathbf{v}_i, \sum_{v_j \in \mathcal{N}_{v_i}} \hat{\mathbf{e}}_{ij}\Big)$, where $v_j \in \mathcal{N}_{v_i}$ denotes a neighboring node of $v_i$ in the graph.

To differentiate the contributions of neighboring nodes, the AMP layer employs learnable parameters $\phi^w$ to predict the anisotropic importance weight of edge feature $\hat{\mathbf{e}}_{ij}$ with respect to node $v_i$. These weights are then normalized across the neighborhood of $v_i$ using a softmax function:

$$w_{ij} = \phi^w(\mathbf{e}_{ij}, \mathbf{v}_i, \mathbf{v}_j), \alpha_{ij} = \frac{\exp(w_{ij})}{\sum_{k \in \mathcal{N}_i} \exp(w_{ik})}. \quad (3)$$

The normalized coefficients are used to compute a linear combination of the corresponding edge features. This linear combination serves as the final input for the node update function $\phi^v$ given node feature $\mathbf{v}_i$:

$$\hat{\mathbf{v}}_i = \phi^v\Big(\mathbf{v}_i, \sum_{v_j \in \mathcal{N}_{v_i}} \alpha_{ij} \hat{\mathbf{e}}_{ij}\Big). \quad (4)$$

The proposed AMP layer enables the implicit assignment of varying contribution weights to the updated edge features within the same neighborhood. Analyzing the learned direction-specific weights in AMP further enhances interpretability. We adopt an MLP implementation for $\phi^w$, while alternative designs, such as graph attention (Veličković et al., 2018) or cross-attention (Vaswani et al., 2017), are also feasible. A detailed comparison is provided in Appendix C.2.

### 3.2. Differentiable Multi-Scale Graph Construction

With the AMP layer functioning within each graph level, local dependencies are effectively propagated throughout the high-resolution graphs, guiding the selection of nodes to be discarded in the next hierarchy for improved long-range modeling. We now delve into the details of the differentiable node selection method (DiffSELECT) for hierarchical graph construction.

In the DiffSELECT operation, we train the node update module $\phi^v$ based on anisotropic aggregated edge features to produce a 2-dimensional probability vector $\boldsymbol{\pi}_i^l$ for each node $v_i$. This vector $\boldsymbol{\pi}_i^l = (\pi_{i,0}^l, \pi_{i,1}^l)$ represents the probabilities of discarding or retaining node $v_i$ in the next-level coarser graph $\mathcal{G}_{l+1}$. We rewrite Eq. (4) as follows:

$$\hat{\mathbf{v}}_i^l,\ \boldsymbol{\pi}_i^l = \phi^v\left(\mathbf{v}_i^l, \sum_{v_j \in \mathcal{N}_{v_i}} \alpha_{ij}^l \hat{\mathbf{e}}_{ij}^l\right). \quad (5)$$

In the next step, we apply Gumbel-Softmax sampling (Jang et al., 2017) independently to each node, using the log-probabilities $(\log \pi_{i,0}^l, \log \pi_{i,1}^l)$ as logits. This produces a soft one-hot vector $\mathbf{z}_i^l = (z_{i,0}^l, z_{i,1}^l)$ for each node:

$$\begin{aligned} z_{i,k}^l &= \text{Gumbel-Softmax}\left(\log \pi_{i,0}^l, \log \pi_{i,1}^l\right) \\ &= \frac{\exp\left((\log \pi_{i,k}^l + g_{i,k}^l)/\tau\right)}{\sum_{k'=0}^1 \exp\left((\log \pi_{i,k'}^l + g_{i,k'}^l)/\tau\right)}, \end{aligned} \quad (6)$$

where $g_{i,k}^l$ is Gumbel noise sampled independently for each node, and $\tau$ is the temperature parameter controlling the smoothness of the sampling. In this way, the node set $\mathcal{V}_{l+1}$ is adaptively constructed based on node features from the finer graph level. The straight-through Gumbel-Softmax estimator provides a differentiable approximation to hard sampling, thereby facilitating end-to-end training. We implement the Gumbel-Softmax with temperature annealing to stabilize training, initially encouraging the exploration of hierarchies and gradually refining the selection process.

The edges $\mathcal{E}_{l+1}$ in the coarser graph $\mathcal{G}_{l+1}$ are constructed by connecting the selected nodes using the original graph's edges $\mathcal{E}_l$. However, this process may result in disconnected partitions (see Appendix Figure 6). To address this issue,

we enhance the connectivity in $\mathcal{E}_{l+1}$ by incorporating the $K$-hop edges during edge selection, defined as follows:

$$\begin{aligned} \widetilde{\mathcal{E}}_l^{(K)} = \mathcal{E}_l \cup \big\{ e_{ij} \mid\ &\exists v_{k_1}, v_{k_2}, \ldots, v_{k_{K-1}} \in \mathcal{V}_l \\ &\text{s.t. } e_{i,k_1}, e_{k_1,k_2}, \ldots, e_{k_{K-1},j} \in \mathcal{E}_l \big\}. \end{aligned} \quad (7)$$

In essence, $e_{ij} \in \widetilde{\mathcal{E}}_l^K$ if there exists a sequence of intermediate nodes $\{v_{k_1}, v_{k_2}, \ldots, v_{k_{K-1}}\}$ consecutively connected by edges in $\mathcal{E}_l$ or $e_{ij} \in \mathcal{E}_l$. The edges in $\mathcal{E}_{l+1}$ are defined as:

$$\mathcal{E}_{l+1} = \big\{ e_{ij} \mid \exists v_i, v_j \in \mathcal{V}_{l+1} \text{ s.t. } e_{ij} \in \widetilde{\mathcal{E}}_l^{(K)} \big\}. \quad (8)$$

$\mathcal{E}_{l+1}$ consists of edges from the enhanced edge set $\widetilde{\mathcal{E}}_l^{(K)}$ that connect nodes in $\mathcal{V}_{l+1}$. As $K$ increases, nodes in $\widetilde{\mathcal{E}}_l^{(K)}$ can be connected through additional intermediate nodes, thereby improving long-range connectivity. In practice, the most effective value of $K$ is found to be 2. We include further discussions in Appendix C.3.

The graph construction process is fully differentiable, allowing for seamless integration into differentiable physical simulators. By flexibly adapting graph hierarchies based on simulation states, it paves the way for more accurate predictions of the spatiotemporal patterns in complex systems.

### 3.3. Inter-Level Feature Propagation with AMP

During the downsampling process from $\mathcal{G}_l$ to the generated coarser graph $\mathcal{G}_{l+1}$, as illustrated in Figure 1, the REDUCE operation aggregates information to each node in $\mathcal{V}_{l+1}$ from its corresponding neighbors in $\mathcal{V}_l$. Conversely, the EXPAND operation unpools the reduced graph back to a finer resolution, delivering the information of the pooled nodes to their neighbors at the finer level.

Prior work employed non-parametric aggregation in inter-level propagation, convolving features based on the normalized node degree. It simplifies intricate relationships between nodes and neglects the directional aspects of information flow. To address this, EvoMesh is designed to learns inter-level aggregation weights that are both data-specific and time-varying. Specifically, the importance weight $\alpha_{ij}^l$ computed by the AMP layer inherently captures the relevance of node $v_j$'s features to node $v_i$ at the graph level $l$. These weights can be directly reused for the REDUCE and EXPAND operations in the downsampling and upsampling processes. We provide details of these operations as follows:

- REDUCE: Let $v_i$ be the node at the coarser graph level. The downsampling process aggregates the information of the current neighbors $\mathcal{N}_i$ by reusing the weight $\alpha_{ij}^l$: $\mathbf{v}_i^{l+1} \leftarrow \text{REDUCE}(\{\mathbf{v}_j^l, \alpha_{ij}^l\}_{j \in \mathcal{N}_i}) := \sum_{j \in \mathcal{N}_i} \alpha_{ij}^l \mathbf{v}_j^l.$

- EXPAND: We first unpool the node features from the coarser graph $\mathcal{G}_{l+1}$ back to the finer level $\mathcal{G}_l$. To achieve

*Table 2.* **Quantitative comparison of the one-step and long-term prediction errors.** We report the mean results over 3 random seeds, with corresponding standard deviations detailed in Appendix C.7. *Promotion* denotes the improvement over the second-best model.

| Model | RMSE-1 ($\times 10^{-2}$) | | | | RMSE-All ($\times 10^{-2}$) | | | |
|---|---|---|---|---|---|---|---|---|
| | Cylinder | Airfoil | Flag | Plate | Cylinder | Airfoil | Flag | Plate |
| MGN (2021) | 0.3046 | 77.38 | 0.3459 | 0.0579 | 59.78 | 2816 | 115.3 | 3.982 |
| Lino *et al.* (2022) | 3.9352 | 85.66 | 0.9993 | 0.0291 | 27.60 | 2080 | 118.2 | 2.090 |
| BSMS-GNN (2023) | 0.2263 | 71.69 | 0.5080 | 0.0632 | 16.98 | 2493 | 168.1 | 1.811 |
| Eagle (2023) | 0.1733 | 51.55 | 0.3805 | 0.0392 | 20.05 | 2344 | 127.7 | 7.797 |
| HCMT (2024) | 0.9190 | 48.62 | 0.4013 | 0.0295 | 23.59 | 3238 | 90.32 | 2.468 |
| EvoMesh | **0.1568** | **41.41** | **0.3049** | **0.0282** | **6.571** | **2002** | **76.16** | **1.296** |
| *Promotion* | **9.53%** | **14.8%** | **11.9%** | **3.10%** | **61.3%** | **3.75%** | **15.7%** | **28.5%** |

this, we record the nodes selected during the downsampling process and use this information to place the nodes back in their original positions in the graph. Then, we reuse the previously computed importance weights $\alpha_{ij}^l$ to assign weighted features from $\mathcal{G}_{l+1}$ back to $\mathcal{G}_l$. The EXPAND operation is formally defined as: $\tilde{\mathbf{v}}_i^l \leftarrow$ EXPAND$(\{\mathbf{v}_j^{l+1}, \alpha_{ij}^l\}_{j \in \mathcal{N}_i}) := \sum_{j \in \mathcal{N}_i} \mathbf{v}_j^{l+1} \alpha_{ij}^l$.

- FeatureMixing: While the EXPAND operation upsamples coarser-level features of $G_{l+1}$ to match the resolution of the current level $G_l$, naïvely upsampled features may suffer from artifacts or misalignment. To mitigate this, we introduce FeatureMixing to refine and fuse coarse-level features with intra-level information at the current resolution. Specifically, we apply an additional anisotropic message passing step to the upsampled features $\tilde{\mathbf{v}}_i^l$ and then integrate these features with the original intra-level features $\mathbf{v}_l$ in $\mathcal{G}_l$ (prior to downsampling) using a skip connection: $\bar{\mathbf{v}}_i^l \leftarrow$ FeatureMixing$(\tilde{\mathbf{v}}_i^l, \mathbf{v}_i^l, \{\mathbf{e}_{ij}^l\}_{j \in \mathcal{N}_i}) := \mathbf{v}_i^l +$ AMP$(\tilde{\mathbf{v}}_i^l, \{\tilde{\mathbf{v}}_j^l\}_{j \in \mathcal{N}_i}, \{\mathbf{e}_{ij}^l\}_{j \in \mathcal{N}_i})$.

### 3.4. Implementation Details

We train EvoMesh using the one-step supervision that measures the $L_2$ loss between the ground truth and the next-step predictions. We include detailed descriptions of the implementation of encoder, decoder, node update function and edge update function in Appendix B.

## 4. Experiments

In this section, we present the key results of the proposed method. Additional analyses can be found in Appendix C.

### 4.1. Experimental Setup

We evaluate EvoMesh on five mesh-based benchmarks from previous work (Pfaff et al., 2021; Cao et al., 2023; Wu et al., 2023; Narain et al., 2012). For detailed descriptions, including the input physical quantities, please refer to Appendix A.

- *CylinderFlow*: Simulation of incompressible flow around a cylinder based on 2D Eulerian meshes.

- *Airfoil*: Aerodynamic simulation around airfoil cross-sections based on 2D Eulerian meshes.

- *FlyingFlag*: Simulation of flag dynamics in the wind based on Lagrangian meshes with fixed topology.

- *DeformingPlate*: Deformation of hyper-elastic plates based on Lagrange tetrahedral meshes.

- *FoldingPaper*: Deformation of paper sheets with evolving meshes driven by varying forces at the four corners.

We mainly compare EvoMesh with the following methods:

- MGN (Pfaff et al., 2021), which performs multiple times of message passing on the original graph.

- Lino *et al.* (Lino et al., 2022), which also trains MPNNs on manually-set multi-scale mesh graphs.

- BSMS-GNN (Cao et al., 2023), which generates static hierarchies using bi-stride pooling and performs message passing on predefined meshes.

- Eagle (Janny et al., 2023), which constructs a two-scale hierarchy using precomputed geometric clustering and performs message passing at both levels.

- HCMT (Yu et al., 2024), which generates static hierarchies by applying Delaunay triangulation to the bi-stride pooled nodes, and enables directed feature propagation with the attention mechanism.

All models are trained using the Adam optimizer for $1M$ steps, with an exponential learning rate decay from $10^{-4}$ to $10^{-6}$ over the first $500K$ steps. We provide further details on the architecture and hyperparameters of the compared models in Appendix D.

### 4.2. Main Results

**Standard benchmarks.** Table 2 presents the root mean squared error (RMSE) of one-step prediction (RMSE-1) and

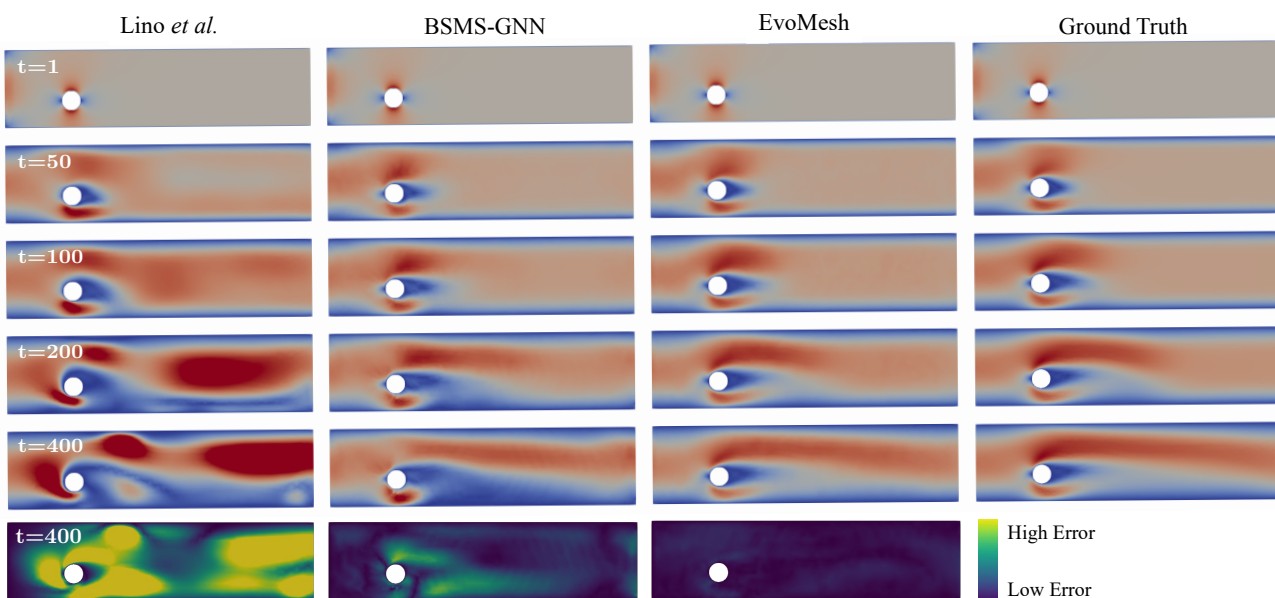

*Figure 2.* **Prediction showcases over 400 future steps on CylinderFlow.** From the displayed error maps, it is evident that EvoMesh effectively captures long-term dynamics, providing predictions that closely align with the ground truth.

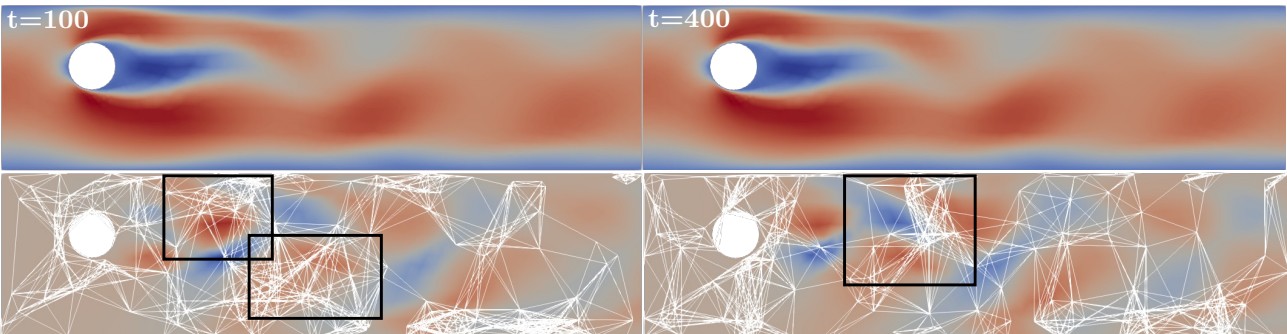

*Figure 3.* **A demonstration of how the learned hierarchies adapt to evolving physical dynamics. Top:** the velocity field from the true data. **Bottom:** the temporal difference of the velocity fields between adjacent time steps alongside the constructed coarser-level mesh graph ($\mathcal{G}_{l=4}$). The highlighted areas demonstrate a notable experimental phenomenon: the mesh dynamically evolves with the data context and aligns with the critical areas of change in the data.

long-term rollouts for 100–600 future time steps (RMSE-all). EvoMesh consistently outperforms the compared models across all benchmarks. This demonstrates the effectiveness of building context-aware, time-evolving hierarchies with learnable, directionally non-uniform feature propagation both within and across graph levels. Figure 2 presents long-term predictions on CylinderFlow, based solely on the system's initial conditions at the first step. As we can see, EvoMesh captures the complex, time-varying fluid flow around the cylinder obstacle more successfully, with its predictions closely matching the ground truth evolution. More results are shown in Appendix C.9.

**Can the learned hierarchies adapt to evolving data dynamics?** In Figure 3, we visualize the time-evolving hierarchies constructed by EvoMesh at different time steps, where coarser-level nodes tend to concentrate in regions

highlighted by the temporal differences in the true data. We have two observations here: First, the constructed hierarchy evolves as the data context changes. Second, the time-evolving graph structures align with the high-intensity regions, either in the velocity fields (top) or in their temporal variations (bottom). These results highlight the effectiveness of our approach in capturing significant dynamic patterns.

**Paper simulation with changing meshes.** We evaluate EvoMesh in a more challenging setting with time-varying meshes for paper folding simulation, generated using the ARCSim solver (Narain et al., 2012; Wu et al., 2023), and compare EvoMesh with MGN (Pfaff et al., 2021). Lino *et al.*, BSMS-GNN, Eagle, and HCMT rely on pre-computed hierarchies during preprocessing, which limits their applicability in scenarios with dynamically changing mesh topologies. Therefore, we do not include them in this evaluation.

*Table 3.* **Simulation results of 2D paper folding with time-varying input meshes.** We here compare EvoMesh with MGN, as BSMS and HCMT require pre-computed hierarchies, which are unsuitable for scenarios involving continuously changing meshes.

| Model | RMSE-1 ($\times 10^{-2}$) | RMSE-All ($\times 10^{-2}$) |
|---|---|---|
| MGN (2021) | 0.0618 | 24.08 |
| EvoMesh | **0.0544** | **7.412** |
| *Promotion* | **12.0%** | **69.2%** |

We assess the models using ground-truth remeshing nodes provided by the ARCSim Adaptive Remeshing component, following the setup from (Pfaff et al., 2021). As shown in Table 3, EvoMesh achieves superior short-term and long-term accuracy compared to MGN, indicating that the time-evolving graph hierarchies in our approach can better fit physical systems with significant geometric variations, as represented by the time-varying input mesh structures.

**Model stability under variable graph structures.** Due to the stochasticity of Gumbel-Softmax sampling in DiffSELECT, we evaluate the stability of trained EvoMesh by conducting three independent runs on the test set. The mean and standard deviations of the prediction errors reveal minimal discrepancies across different runs, as shown in Table 11 in Appendix C.6. These findings demonstrate that once trained, EvoMesh generates consistent graph hierarchies based on the same inputs.

### 4.3. Ablation Studies

EvoMesh has three key components: *(i)* evolving graph hierarchy, *(ii)* anisotropic intra-level propagation, *(iii)* learnable inter-level propagation. To evaluate the contribution of each component, we implement several ablated variants of EvoMesh, including: *Static(Bi-stride)-Anisotropic-Unlearnable* (*M1*), *Static(Bi-stride)-Anisotropic-Learnable* (*M2*), *Uniform-Anisotropic-Learnable* (*M3*), and *Dynamic-Anisotropic-Unlearnable* (*M4*), and compare them against the BSMS-GNN baseline, which uses static hierarchies, isotropic intra-level summation, and unlearnable inter-level propagation. Both *M1* and *M2* adopt the same static bi-stride hierarchy via preprocessing as BSMS-GNN. *M3* constructs the hierarchy via uniform node sampling in each hierarchy, while *M4* applies dynamic hierarchy construction without learnable inter-level updates.

Figure 4 demonstrates the effectiveness of direction-aware message propagation at both intra- and inter-levels, as well as the benefit of learning dynamic graph hierarchies. Comparing *M1* with BSMS-GNN shows that integrating AMP improves performance even under a static hierarchy. Furthermore, the comparison between EvoMesh and *M4*, as well as between *M1* with *M2*, highlights the importance of learnable inter-level propagation in capturing hierarchical signal flow. In addition, although *M3* benefits from learnable

propagation, its use of uniform node sampling results in sub-optimal performance, suggesting the necessity of adaptive and data-aware hierarchy construction.

Furthermore, in Figure 5, we visualize the variance of predicted anisotropic edge weights and compare it with areas where physical quantities present substantial variations over time. The results reveal a strong correlation between the anisotropic learning mechanism and the rapidly changing dynamics of the physical system.

### 4.4. Generalization Analyses

**Generalization to out-of-distribution mesh resolutions.** Nearly all existing machine learning models for mesh-based simulations are not resolution-free and may fail when evaluated on unseen mesh resolutions. We assess the generalization performance of EvoMesh by training it on low-resolution meshes and testing it on high-resolution meshes. The average number of nodes in the test data is twice that of the training data, and the number of edges is three times greater. Table 4 reports one-step and 50-step rollout errors on out-of-distribution (OOD) mesh resolutions. EvoMesh shows strong generalization on the CylinderFlow and Airfoil datasets, highlighting its zero-shot capability to handle refined mesh structures. This performance gain is largely attributed to EvoMesh's ability to construct hierarchical graphs adaptively, allowing it to scale effectively with increased resolution. On the other hand, MGN achieves lower errors on the FlyingFlag and DeformingPlate datasets, indicating that mesh-based architectures remain effective for systems with more regular structures and smoother deformation dynamics. While our method does not yet achieve full generalization across arbitrary resolutions, truly resolution-free modeling remains an open challenge that calls for more advanced architectural design. Nevertheless, this holds significant value in practical applications and has the potential to greatly reduce the time overhead of numerical simulation processes for preparing the large-scale mesh data required for model training.

**Generalization to physical variations.** We evaluate EvoMesh under strong distribution shifts in the input physical quantities. Table 5 presents data statistics and the RMSE results on the CylinderFlow and Airfoil datasets. EvoMesh consistently outperforms the compared models in both short-term and long-term simulations. This advantage mainly comes from its ability to learn evolving hierarchies and model intra-level and inter-level interactions based on physical context. When the fluid dynamics in the test set become more complex—characterized by increased variance in the velocity field over time—the dynamics patterns propagate more rapidly in space. EvoMesh adaptively constructs graph hierarchies and more effectively captures long-range node interactions in response to evolving physical contexts.

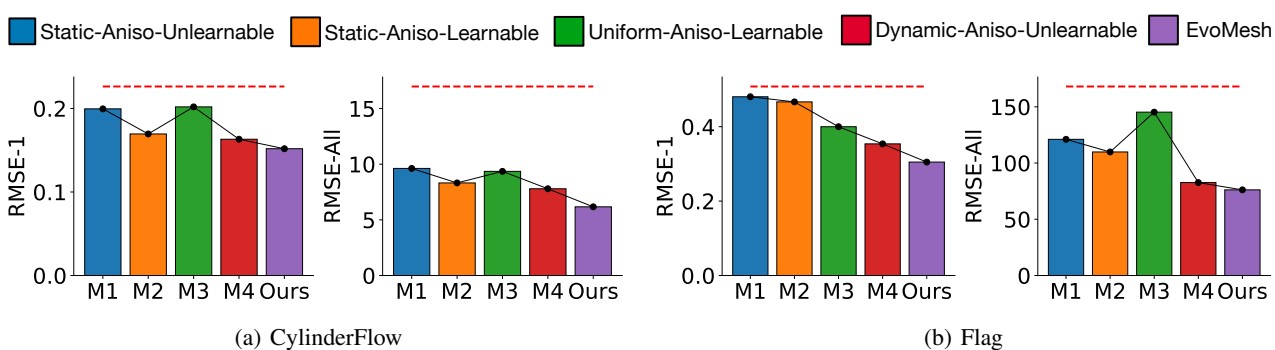

*Figure 4.* **Ablation studies.** We provide analyses of time-evolving hierarchies, anisotropic intra-level propagation, and learnable inter-level feature propagation. The red dashed lines represent results from BSMS-GNN (Cao et al., 2023). Lower values indicate better performance.

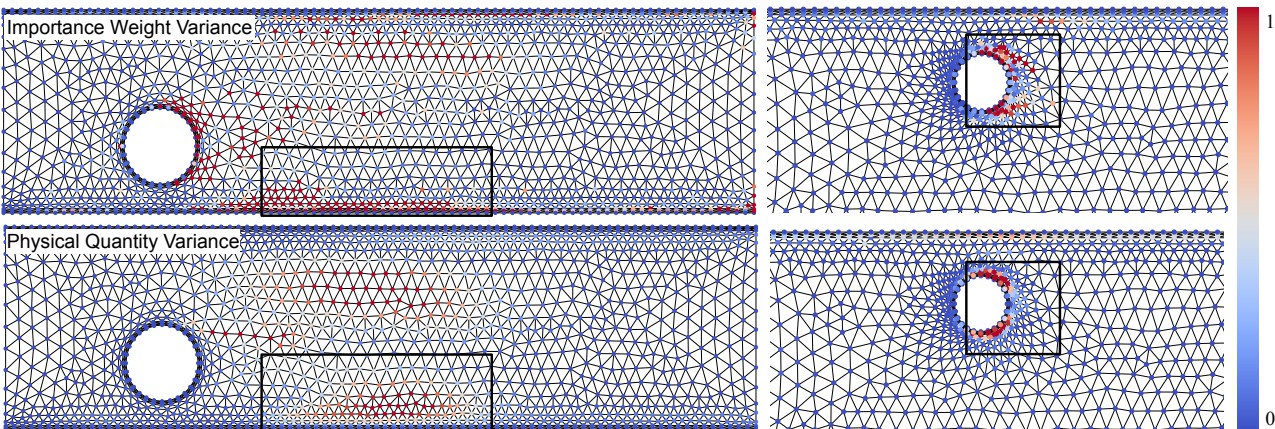

*Figure 5.* **A demonstration of how the predicted anisotropic edge weights respond to dramatic changes in physical quantities over time. Top:** Visualizations of the variance in the generated anisotropic weights, calculated on adjacent edges. **Bottom:** Variance in physical quantities over time. The strong correlation between them highlights the AMP's ability to detect significant patterns in data.

## 5. Related Work

**Learning-based and GNN-based physical simulation.**
Recent advances have demonstrated that learning-based approaches can efficiently tackle complex and high-dimensional physical simulation tasks, including fluid dynamics (Zhu et al., 2024), structural analysis (Kavvas et al., 2018; Thai, 2022), and climate modeling (Kurth et al., 2018; Rasp et al., 2018; Rolnick et al., 2022; Lam et al., 2023). These methods can be broadly categorized by their data representations: partial differential equations (Raissi et al., 2017; 2019; Lu et al., 2019; Li et al., 2021; Wang et al., 2021), particle-based systems (Li et al., 2019; Sanchez-Gonzalez et al., 2020; Ummenhofer et al., 2020; Prantl et al., 2022), and mesh-based systems (Pfaff et al., 2021; Lino et al., 2022; Fortunato et al., 2022; Cao et al., 2023). The rapid inference and differentiable nature of these models have facilitated a range of downstream applications, such as inverse design (Wang & Zhang, 2021; Goodrich et al., 2021; Allen et al., 2022; Janny et al., 2023). In particular, Graph Neural Networks (GNNs) have emerged as a powerful tool for modeling physical systems across various domains, including articulated bodies (Sanchez-Gonzalez

et al., 2018), soft-body deformation and fluids (Li et al., 2019; Mrowca et al., 2018; Sanchez-Gonzalez et al., 2020; Rubanova et al., 2022; Wu et al., 2023), rigid body dynamics (Battaglia et al., 2016; Li et al., 2019; Mrowca et al., 2018; Bear et al., 2021; Rubanova et al., 2022), and aerodynamics (Belbute-Peres et al., 2020; Hines & Bekemeyer, 2023; Pfaff et al., 2021; Fortunato et al., 2022; Cao et al., 2023). Among these, MeshGraphNets (Pfaff et al., 2021) serves as a representative, introducing a general scheme for representing meshes as graphs and learning mesh-based dynamics, inspiring subsequent works that focus on improving modeling capacity and computational efficiency.

**Hierarchical GNNs for physical simulation.** Hierarchical GNNs leverage multi-scale graph structures (Lino et al., 2022; Han et al., 2022; Fortunato et al., 2022; Allen et al., 2023; Janny et al., 2023; Cao et al., 2023; Yu et al., 2024) to reduce computational overhead by operating on coarser representations and to facilitate long-range information propagation. For example, GMR-Transformer-GMUS (Han et al., 2022) employs uniform sampling for pooling, while Eagle (Janny et al., 2023) adopts a two-scale message passing scheme with geometric clustering by preprocessing the

*Table 4.* **One-step (RMSE-1) and 50-step (RMSE-50) rollout prediction errors on out-of-distribution (OOD) mesh resolutions.** The average number of nodes in the test data is twice that of the training data, and the number of edges is three times greater.

| Model | RMSE-1 ($\times 10^{-2}$) | | | | RMSE-50 ($\times 10^{-2}$) | | | |
|---|---|---|---|---|---|---|---|---|
| | Cylinder | Airfoil | Flag | Plate | Cylinder | Airfoil | Flag | Plate |
| MGN (2021) | 1.0596 | 169.6 | **0.4215** | 0.0359 | 7.833 | 1829 | **55.96** | **0.2467** |
| Lino *et al.* (2022) | 25.893 | 144.4 | 0.8906 | 0.0475 | 65.21 | 1391 | 93.68 | 3.9845 |
| BSMS-GNN (2023) | 0.9177 | 202.3 | 0.6486 | 0.0474 | 2.097 | 1677 | 59.18 | 0.2554 |
| HCMT (2024) | 1.3864 | 205.5 | 1.0634 | **0.0354** | 7.541 | 2569 | 86.87 | 0.2957 |
| EvoMesh | **0.4855** | **126.7** | 0.5536 | 0.0368 | **1.077** | **812.5** | 58.29 | 0.3780 |

*Table 5.* **Generalization results across domains with various scales of input velocities.** The domain gap is presented by the variance and norm of data in training/test splits. *Increase* denotes the relative increase of the test data compared to the training data.

| Split | Cylinder | | Airfoil | |
|---|---|---|---|---|
| | Var | Norm | Var | Norm |
| Train | 7.92 | 579.6 | 288.3 | 173.4 |
| Test | 13.43 | 826.3 | 827.4 | 180.6 |
| *Increase* | 64.5% | 42.5% | 186.9% | 4.20% |

| Model | Cylinder | | Airfoil | |
|---|---|---|---|---|
| | RMSE-1 | RMSE-All | RMSE-1 | RMSE-All |
| MGN | $4.99 \times 10^{-3}$ | 1.020 | 1.193 | 88.23 |
| Lino *et al.* | $5.60 \times 10^{-3}$ | 1.415 | 3.226 | 410.5 |
| BSMS-GNN | $2.58 \times 10^{-3}$ | 0.251 | 1.035 | 30.32 |
| Eagle | $2.49 \times 10^{-3}$ | 0.273 | 0.931 | 51.83 |
| HCMT | $7.35 \times 10^{-3}$ | 1.047 | 1.697 | 63.18 |
| EvoMesh | **$2.14 \times 10^{-3}$** | **0.091** | **0.665** | **22.57** |
| *Promotion* | **14.1%** | **63.7%** | **28.6%** | **25.6%** |

mesh structure. LayersNet (Shao et al., 2023) introduces a static, patch-based hierarchy for garment animation, simplifying interaction modeling via particle patches. More recent works (Lino et al., 2022; Cao et al., 2023; Yu et al., 2024; Garnier et al., 2024; Hy & Kondor, 2023) integrate hierarchical GNNs with U-Net architectures (Ronneberger et al., 2015), using static multi-level structures for message passing. For instance, Lino et al. (2022) relies on manually defined grid resolutions, BSMS-GNN (Cao et al., 2023) proposes a bi-stride pooling strategy with enhanced edge connectivity, and HCMT (Yu et al., 2024) further refines the hierarchy using Delaunay triangulation and replaces message passing with graph attention. However, these methods typically employ precomputed, static hierarchies and uniform feature aggregation, limiting their adaptability to dynamic physical environments. In contrast, our approach constructs context-aware, temporally evolving graph hierarchies with learnable anisotropic feature propagation, enabling robust adaptation to diverse initial conditions and rapidly changing dynamics.

**Differentiable graph pooling.** A variety of differentiable and learnable graph pooling methods have been proposed to enable end-to-end training of hierarchical graph representa-

tions, such as DiffPool (Ying et al., 2018), TopKPool (Gao & Ji, 2019), SAGPool (Lee et al., 2019), and ASAPooling (Ranjan et al., 2020). These methods construct hierarchical representations of graphs by either learning soft cluster assignments or selecting top-ranked nodes based on learned importance scores. However, most of these methods are primarily designed for static graphs and focus on global graph-level tasks, where unpooling or reconstruction of the original graph structure is not required. In contrast, mesh-based physical simulation demands fine-grained local information and temporally-evolving hierarchies for accurate modeling of physical dynamics. Our approach differs by constructing context-aware, time-varying graph hierarchies tailored for mesh-based simulation, enabling flexible integration of global and local features and supporting dynamic adaptation to changing physical conditions.

## 6. Conclusions and Limitations

In this paper, we introduced EvoMesh, a neural network that significantly advances the state-of-the-art in mesh-based simulation. Our key innovation is adaptively creating the temporally-evolving and context-aware graph structures of hierarchical GNNs through a differentiable node selection process. To this end, we proposed an anisotropic message passing mechanism to enhance the propagation of long-term dependencies between distant nodes, aligning with the directed nature of significant dynamic patterns. Extensive experiments show that EvoMesh outperforms existing models, especially those with fixed graph hierarchies, in both short-term and long-term predictions.

A potential limitation of this work is the need to improve the interpretability of the learned hierarchies. Additionally, we would consider incorporating physical priors into EvoMesh to enhance the model's robustness and generalizability, particularly in *resolution-free* problem settings, which have been less explored in existing mesh-based approaches.

## Acknowledgments

This work was supported by the National Natural Science Foundation of China (62250062), the Smart Grid National Science and Technology Major Project (2024ZD0801200),

the Shanghai Municipal Science and Technology Major Project (2021SHZDZX0102), and the Fundamental Research Funds for the Central Universities.

## Impact Statement

In this work, we adhere to the highest ethical standards throughout all stages of research. No human subjects were involved, and no personal data was used, ensuring full compliance with privacy and security protocols. All datasets employed are publicly available, minimizing concerns regarding sensitive information exposure.

We recognize that while physical simulation models can drive significant advancements, they also have the potential for misuse if applied irresponsibly. Thus, we emphasize the importance of thoughtful consideration regarding the context and ethical implications when deploying these models, particularly in high-stakes applications such as healthcare, engineering, and environmental management.

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

# Appendix

## A. Datasets

We employ four established datasets from MGN (Pfaff et al., 2021): *CylinderFlow*, *Airfoil*, *Flag*, and *DeformingPlate*.

- The *CylinderFlow* case examines the transient incompressible flow field around a fixed cylinder positioned at different locations, with varying inflow velocities.
- The *Airfoil* case explores the transient compressible flow field at varying Mach numbers around the airfoil, with different angles of attack.
- The *Flag* case involves a flag blowing in the wind on a fixed Lagrangian mesh.
- The *DeformingPlate* case involves hyperelastic plates being compressed by moving obstacles.

The *CylinderFlow*, *Airfoil*, and *Flag* datasets are each split into $1,000$ training sequences, $100$ validation sequences, and $100$ testing sequences. The D*eformingPlate* dataset is split into $500$ training sequences, $100$ validation sequences, and $100$ testing sequences.

We also consider a more challenging dataset, *FoldingPaper*, where varying forces at the four corners deform paper with time-varying Lagrangian mesh graphs, generated using the ARCSim solver (Narain et al., 2012; Wu et al., 2023). This dataset is divided into $500$ training sequences, $100$ validation sequences, and $100$ testing sequences.

We present statistical details of all five datasets in Table 6 and the input physical quantities in Table 7.

*Table 6.* Statistics of the CylinderFlow, Airfoil, Flag, DeformingPlate, and FoldingPaper datasets.

| Dataset | Average # nodes | Average # edges | Mesh type | # Hierarchies | # Steps |
|---|---|---|---|---|---|
| CylinderFlow | 1886 | 5424 | triangle, 2D | 7 | 600 |
| Airfoil | 5233 | 15449 | triangle, 2D | 7 | 100 |
| Flag | 1579 | 9212 | triangle, 2D | 7 | 400 |
| DeformingPlate | 1271 | 4611 | tetrahedron, 3D | 6 | 400 |
| FoldingPaper | 110 | 724 | triangle, 2D | 3 | 325 |

*Table 7.* Comparisons of the edge offsets and node inputs of different physical systems.

| Dataset | Type | Edge offset $\mathbf{e}_{ij}$ | Node Input $\mathbf{v}_i$ | Outputs | Noise Scale |
|---|---|---|---|---|---|
| CylinderFlow | Eulerian | $X_{ij}, \lvert X_{ij} \rvert$ | $v_i, n_i$ | $\dot{v}_i$ | $v_i : 2e-2$ |
| Airfoil | Eulerian | $X_{ij}, \lvert X_{ij} \rvert$ | $\rho_i, v_i, n_i$ | $\dot{v}_i, \dot{\rho}_i, P_i$ | $v_i : 2e-2, \rho_i : 1e1$ |
| Flag | Lagrangian | $X_{ij}, \lvert X_{ij} \rvert, x_{ij}, \lvert x_{ij} \rvert$ | $\dot{x}_i, n_i$ | $\dot{x}_i$ | $x_i : 3e-3$ |
| DeformingPlate | Lagrangian | $X_{ij}, \lvert X_{ij} \rvert, x_{ij}, \lvert x_{ij} \rvert$ | $\dot{x}_i, n_i$ | $\dot{x}_i$ | $x_i : 3e-3$ |
| FoldingPaper | Lagrangian | $X_{ij}, \lvert X_{ij} \rvert, x_{ij}, \lvert x_{ij} \rvert$ | $\dot{x}_i, n_i$ | $\dot{x}_i$ | $x_i : 3e-3$ |

## B. Model Implementation

We present model configurations of different physical systems below:

- **Edge offsets.** $X$ and $x$ stand for the mesh-space and world-space position. For an Eulerian system, only mesh position is used for $\mathbf{e}_{ij}$, while for a Lagrangian system, both mesh-space and world-space positions are used. The edge offsets are directly used as low-dimensional input to the edge update function $\phi^e$. These features are concatenated and fed into $\phi^e$ without any transformation through an MLP or other encoding processes to generate a higher-dimensional representation.

- **Input and target of the physical term of node $v_i$.** $v$ is the velocity, $\rho$ is the density, $P$ is the absolute pressure, and the dot $\dot{a} = a_{t+1} - a_t$ stands for temporal change for a variable $a$. $n$ stands for the node type of $v_i$. Random Gaussian noise is added to the node input features to enhance robustness during training (Pfaff et al., 2021; Sanchez-Gonzalez et al., 2020; Cao et al., 2023). All the variables involved are normalized to zero-mean and unit variance via preprocessing. The preprocessed physical term is fed to the encoder to transform it into a high-dimensional representation.

The encoder, decoder, node update function $\phi^v$, and edge update function $\phi^e$ all utilize two-layer MLPs with ReLU activation and a hidden size of $128$. Similarly, the importance weight network $\phi^w$ in AMP is implemented using a two-layer MLP.

*Table 8.* Qualitative results of model variants of EvoMesh and the baseline model.

| Model | RMSE-1 ($\times 10^{-2}$) | | RMSE-All ($\times 10^{-2}$) | |
|---|---|---|---|---|
| | Cylinder | Flag | Cylinder | Flag |
| BSMS-GNN (Cao et al., 2023) | 0.2263 | 0.5080 | 16.98 | 168.1 |
| Static(Bi-stride)-Anisotropic-Unlearnable (*M1*) | 0.1995 | 0.4804 | 9.621 | 121.1 |
| Static(Bi-stride)-Anisotropic-Learnable (*M2*) | 0.1695 | 0.4666 | 8.317 | 109.9 |
| Uniform-Anisotropic-Learnable (*M3*) | 0.2019 | 0.3999 | 9.357 | 145.27 |
| Dynamic-Anisotropic-Unlearnable (*M4*) | 0.1631 | 0.3538 | 7.793 | 82.65 |
| EvoMesh | **0.1568** | **0.3049** | **6.571** | **76.16** |

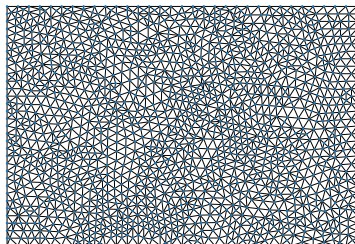 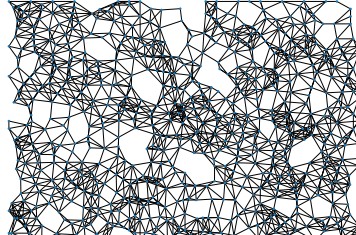 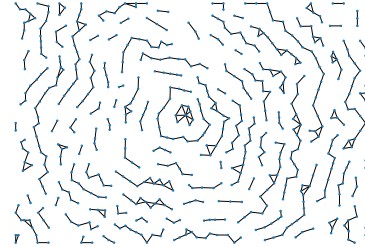

*Figure 6.* Mesh visualization on Flag Dataset. Original mesh (*left*), sub-level graph after differentiable node selection with $K$-hop enhancement with $K = 2$ (*middle*), and sub-level graph after node selection without $K$-hop enhancement (*right*).

LayerNorm is applied to the MLP outputs, except for the decoder and the importance weight network. We set $K = 2$ for edge enhancement, which is aligned with the setting of BSMS-GNN (Cao et al., 2023). In the Gumbel-Softmax for differentiable node selection, temperature annealing decreases the temperature from 5 to 0.1 using a decay factor of $\gamma = 0.999$, encouraging exploration of hierarchies while gradually refining their selection to ensure stability. EvoMesh is trained with Adam optimizer, using an exponential learning rate decay from $10^{-4}$ to $10^{-6}$. All experiments are conducted using 4 Nvidia RTX 3090. We mainly build EvoMesh based on the released code of BSMS-GNN (Cao et al., 2023).

## C. Additional Results

### C.1. Ablation Study

In Sec. 4.3, we compare different variants of our EvoMesh model against the BSMS-GNN baseline, to evaluate the effectiveness of *(i)* dynamic hierarchy construction based on the input mesh topology and physical quantities, *(ii)* anisotropic intra-level feature propagation, *(iii)* learnable inter-level feature propagation. The variants we investigate include:

- *Static(Bi-stride)-Anisotropic-Unlearnable (**M1**)*,
- *Static(Bi-stride)-Anisotropic-Learnable (**M2**)*,
- *Uniform-Anisotropic-Learnable (**M3**)*,
- *Dynamic-Anisotropic-Unlearnable (**M4**)*.

In this ablation study, we utilize a static graph hierarchy preprocessed using bi-stride pooling as described in the BSMS-GNN paper (Cao et al., 2023) for static hierarchy variants, along with a non-parametric intra-level aggregation function from previous works (Pfaff et al., 2021; Cao et al., 2023). Additionally, BSMS-GNN employs unlearnable node degree metrics to generate inter-level aggregation weights, which convolve features based on the normalized node degree for inter-level propagation. We show the quantitative RMSE values of Figure 4 in Table 8.

### C.2. Alternative Implementation of AMP

We evaluate the effectiveness of the AMP module by comparing it with alternative attention-based designs (Veličković et al., 2018; Vaswani et al., 2017).

Unlike standard attention mechanisms such as Graph Attention Networks(GAT) (Veličković et al., 2018), AMP enables differentiable dynamic hierarchy construction—an essential capability for modeling evolving physical systems. While

*Table 9.* Results of different implementations of AMP module.

| AMP Implementation | Cylinder | | Flag | |
|---|---|---|---|---|
| | RMSE-1 ($\times 10^{-2}$) | RMSE-All ($\times 10^{-2}$) | RMSE-1($\times 10^{-2}$) | RMSE-All ($\times 10^{-2}$) |
| EvoMesh-GATConv | 0.2025 | 10.253 | 0.3009 | 106.2 |
| EvoMesh-CrossAttention | 0.1881 | 8.356 | 0.3650 | 75.12 |
| EvoMesh | 0.1568 | 6.571 | 0.3049 | 76.16 |

AMP and GAT both compute importance weights, their usage differs substantially: GAT uses these weights to aggregate node features, whereas AMP directly applies the predicted weights to edge features and reuses them for inter-level feature propagation. This dual mechanism allows AMP to jointly support dynamic hierarchy learning and physics modeling via end-to-end training. To assess the benefits of AMP over GAT, we replace AMP with GATConv (using a single attention head) in EvoMesh. As shown in Table 9, the AMP-based model consistently outperforms the GAT-based variant across datasets, highlighting the importance of dynamic hierarchy construction.

We further explore the flexibility of AMP by implementing it with a cross-attention mechanism (Vaswani et al., 2017) in place of the default MLP. Specifically, edge features $\hat{\mathbf{e}}_{ij}$ are refined by attending to node features $\mathbf{v}_i$ through scaled dot-product attention:

$$\hat{\mathbf{e}}_{\text{aggr}} = \text{CrossAttention}(Q = \mathbf{v}_i, , K = \hat{\mathbf{e}}_{ij}, , V = \hat{\mathbf{e}}_{ij}), \tag{9}$$

and the resulting attention scores are reused for inter-level propagation. Experimental results in Table 9 show that while the cross-attention implementation is competitive, the original MLP-based AMP achieves better overall accuracy. This suggests that the edge features $\hat{\mathbf{e}}_{ij}$ already encode relevant node information, making the explicit inclusion of $\mathbf{v}_i$ through attention redundant and potentially less efficient.

### C.3. Edge Enhancement

When constructing the lower-level graph $\mathcal{G}_{l+1}$ based on the selected nodes, the edges $\mathcal{E}_{l+1}$ are formed by connecting these nodes using the original edges $\mathcal{E}_l$ from the previous graph. However, this approach may lead to disconnected partitions, as observed in previous work (Lee et al., 2019; Cao et al., 2023; Gao & Ji, 2019), and illustrated in Figure 6. To address this issue, we enhance the connectivity of $\mathcal{E}_{l+1}$ by incorporating $K$-hop edges during the edge construction process. We investigate the impact of different $K$ values, specifically $K = 2, 3, 4$, on the Flag dataset. The results are presented in Table 10, along with comparisons of the computational efficiency.

Notably, $K = 2$ yields the lowest RMSE across all conditions (RMSE-1, RMSE-50, and RMSE-all), indicating superior performance compared to higher $K$ values. Despite the performance decline observed with $K = 3$ and $K = 4$, they still outperform the baseline results, indicating the effectiveness of dynamic hierarchical modeling and anisotropy message passing.

*Table 10.* Results for different values of $K$ in edge enhancement. Here, $K = 1$ denotes directly using edges of selected nodes from previous graph levels. Training time and memory usage are measured with a batch size of 32, while inference time and memory are evaluated with a batch size of 1.

| | RMSE-1 ($\times 10^{-2}$) | RMSE-All ($\times 10^{-2}$) | Training | | Infer | |
|---|---|---|---|---|---|---|
| | | | Time (ms) | vRAM (GBs) | Time (ms) | vRAM (GBs) |
| $K = 1$ | 0.3296 | 100.1 | **31.57** | **14.75** | **23.60** | **1.17** |
| $K = 2$ | **0.3049** | **76.16** | 33.67 | 16.53 | 26.33 | 1.24 |
| $K = 3$ | 0.3380 | 86.84 | 34.67 | 18.49 | 33.21 | 1.25 |
| $K = 4$ | 0.3510 | 105.4 | 35.27 | 18.76 | 32.25 | 1.28 |

### C.4. Hyperparameter Analyses on Number of Hierarchies

We conduct an ablation study to assess the impact of varying numbers of hierarchies on model performance. The results from the CylinderFlow dataset, illustrated in Figure 7, demonstrate that EvoMesh consistently outperforms BSMS-GNN across all tested numbers of graph hierarchies. Both models show improved performance with increased hierarchy depth up to 7, indicating that deeper levels help capture more complex interactions and thus enhance accuracy. However, a slight

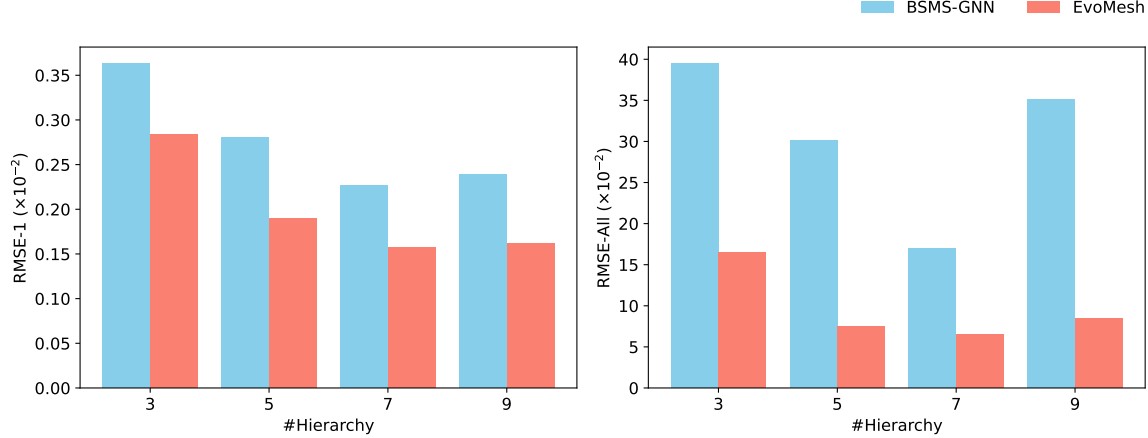

*Figure 7.* Model comparisons on different numbers of graph hierarchies on CylinderFlow dataset.

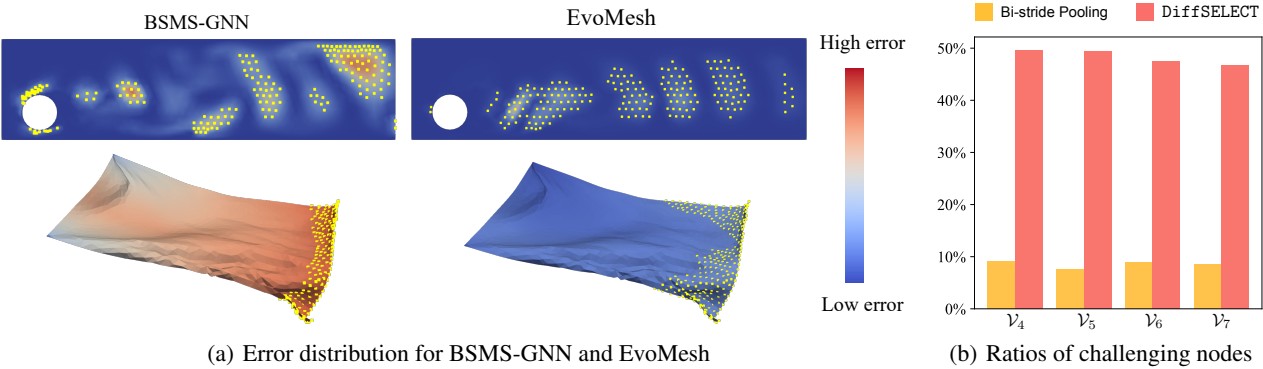

(a) Error distribution for BSMS-GNN and EvoMesh

(b) Ratios of challenging nodes

*Figure 8.* (a) Error maps, where nodes with the top $10\%$ of errors in each model's predictions are marked in yellow and referred to as "*challenging nodes*". (b) EvoMesh retains more challenging nodes in coarser graph hierarchies to capture multi-scale dependencies.

performance decline is observed at level 9, which may suggest the onset of overfitting. Overall, the dynamically learned hierarchies in EvoMesh are shown to be more effective compared to the predefined static hierarchies used in BSMS-GNN.

### C.5. Effectiveness of Evolving Hierarchies

From Figure 4, by comparing EvoMesh vs. *M2* and *M4* vs. *M1*, we observe the advantages of learning adaptive and temporally evolving hierarchical graph structures. These results highlight the significance of adaptively modeling interactions in context-dependent graphs. To better understand how EvoMesh constructs dynamic hierarchies, we visualize the distribution of nodes with the top $10\%$ prediction errors in Figure 8(a). Accordingly in Figure 8(b), we observe that EvoMesh retains a higher proportion of "challenging" nodes in the coarser message passing levels, enabling our model to capture multi-scale dependencies more effectively, especially in areas where finer message passing levels struggle. In contrast, the predefined static hierarchies in the Bi-stride pooling baseline are data-independent and may inevitably overlook modeling long-range relations surrounding these pivotal nodes, even though they typically present higher errors than those in EvoMesh.

### C.6. Stability Analysis

Given the inherent randomness introduced by the Gumbel-Softmax sampling process in `DiffSELECT`, we evaluated the stability of EvoMesh by running the trained model on the test set in three independent trials. We report the mean and standard deviation of the prediction errors in Table 11. Despite the stochastic nature of the node selection process, the results show a very small standard deviation, demonstrating that EvoMesh reliably constructs stable and consistent dynamic hierarchies. This stability can be attributed to the `DiffSELECT` operation, where the node update module $\phi^v$ generates probabilities for retaining nodes in the next-level graph based on anisotropic aggregated edge features. The Gumbel-Softmax

*Table 11.* Evaluation of EvoMesh with three independent tests.

| | Cylinder | Airfoil | Flag | Plate |
|---|---|---|---|---|
| RMSE-1 ($\times 10^{-2}$) | 0.1506 ±3.6E-4 | 36.27 ±5.7E-4 | 0.2741 ±2.4E-2 | 0.0263 ±5.6E-6 |
| RMSE-All ($\times 10^{-2}$) | 6.317 ±0.33 | 2018 ±130 | 68.66 ±2.9 | 1.327 ±0.002 |

*Table 12.* Full quantitative results over three training seeds.

| | Cylinder | Airfoil | Flag | Plate |
|---|---|---|---|---|
| RMSE-1 ($\times 10^{-2}$) | | | | |
| MGN (2021) | 0.3046 ±1.08E-2 | 77.38 ±1.34E+1 | 0.3459 ±6.34E-2 | 0.0579 ±2.64E-3 |
| Lino *et al.* (2022) | 3.9352 ±11.3E-2 | 85.66 ±0.35E+1 | 0.9993 ±2.44E-2 | 0.0291 ±0.19E-3 |
| BSMS-GNN (2023) | 0.2263 ±4.39E-2 | 71.69 ±1.41E+1 | 0.5080 ±0.48E-2 | 0.0632 ±14.3E-3 |
| Eagle (2023) | 0.1733 ±3.02E-2 | 0.385 ±1.03E+1 | 51.55 ±0.87E-2 | 0.0392 ±0.52E-3 |
| HCMT (2024) | 0.9190 ±61.2E-2 | 48.62 ±0.51E+1 | 0.4013 ±1.76E-2 | 0.0295 ±3.45E-3 |
| EvoMesh | **0.1568 ±0.94E-2** | **41.41 ±0.66E+1** | **0.3049 ±6.34E-2** | **0.0282 ±2.65E-3** |
| RMSE-All ($\times 10^{-2}$) | | | | |
| MGN (2021) | 59.78 ±2.00E+1 | 2816 ±1.99E+2 | 115.3 ±1.30E+1 | 3.982 ±1.14E-2 |
| Lino *et al.* (2022) | 27.60 ±0.86E+1 | 2080 ±0.39E+2 | 118.2 ±0.58E+1 | 2.090 ±13.2E-2 |
| BSMS-GNN (2023) | 16.98 ±0.12E+1 | 2493 ±1.70E+2 | 168.1 ±0.65E+1 | 1.811 ±0.42E-2 |
| Eagle (2023) | 20.05 ±0.67E+1 | 2344 ±2.11E+2 | 127.7 ±0.88E+1 | 7.797 ±2.35E-2 |
| HCMT (2024) | 23.59 ±1.38E+1 | 3238 ±3.62E+2 | 90.32 ±0.50E+1 | 2.468 ±42.4E-2 |
| EvoMesh | **6.571 ±0.06E+1** | **2002 ±1.02E+2** | **76.16 ±1.30E+1** | **1.296 ±1.14E-2** |

technique, coupled with temperature annealing, enables differentiable and stable node selection across hierarchy levels. As a result, the dynamic hierarchies are constructed in a manner that is not only consistent but also optimized for long-range dependencies. Moreover, the prediction errors from EvoMesh are significantly smaller than those of the baseline models, underscoring the robustness and reliability of the model, even with its dynamic node selection mechanism.

### C.7. Full Results over Multiple Training Seeds

In Table 2 in the main manuscript, we report the mean results calculated over three random seeds. In Table 12, we provide full comparisons between our model and the baseline models, including standard deviations.

### C.8. Computation Efficiency

We evaluate computational efficiency based on four criteria: training hours, inference time per step, and the total number of model parameters. The results are presented in Table 13.

### C.9. Rollout Showcases

Figures 9 showcase rollout error maps for the Airfoil, Flag, and DeformingPlate datasets. EvoMesh exhibits much lower rollout errors than the baseline models.

### C.10. Constructed Dynamic Hierarchies

We visualize the constructed context-aware and temporally evolving hierarchies in Figure 10. We can see that the constructed hierarchies evolve as the input context changes and the evolving graph structures align with high-intensity regions. More visualizations of the evolution of the graph structure across the sequence are included in the supplementary material.

## D. Baseline Details

We compare EvoMesh with following competitive baselines: (1) MGN (Pfaff et al., 2021) which performs multiple message passing on the input high-resolution mesh topology; (2) Lino *et al.*(Lino et al., 2022), which uses manually set grid resolutions and spatial proximity for graph pooling; (3) BSMS-GNN (Cao et al., 2023), which uses predefined bi-stride pooling prior as preprocessing to generate static hierarchies on same mesh topology; (4) Eagle (Yu et al., 2024), which uses

*Table 13.* The detailed measurements of computation efficiency for EvoMesh and baseline models.

| Measurements | Dataset | BSMS-GNN | HCMT | EvoMesh |
|---|---|---|---|---|
| | Cylinder | 37.11 | 80.60 | 35.96 |
| Training cost (hrs) | Airfoil | 79.09 | 114.32 | 75.45 |
| | Flag | 18.27 | 66.70 | 17.14 |
| | Plate | 39.80 | 99.84 | 41.85 |
| | Cylinder | 16.55 | 79.52 | 21.79 |
| Infer time/step (ms) | Airfoil | 38.04 | 106.34 | 58.84 |
| | Flag | 17.18 | 85.87 | 26.33 |
| | Plate | 28.44 | 100.78 | 47.45 |
| | Cylinder | 2.05M | 2.03M | 2.66M |
| # Parameter | Airfoil | 2.58M | 2.03M | 2.27M |
| | Flag | 2.06M | 2.03M | 2.67M |
| | Plate | 2.87M | 2.03M | 3.20M |

a two-scale hierarchical message-passing approach, downscaling mesh resolution via geometric clustering of mesh positions; (5) HCMT (Yu et al., 2024), which uses Delauny triangulation based on bi-stride nodes and adopt attention mechanism to enable non-uniform feature propagation. The architecture details of the compared models are as follows:

- **MGN.**    In MGN, we use 15 message passing steps in all datasets. The encoder, decoder, node update function, and edge update function are configured in the same way as in our model.

- **Lino *et al.***    We use the four-scale GNN structure proposed in the work of Lino et al. (2022). The edge length of the smallest cell for each dataset is $1/10$ of the average scene size, with each lower scale doubling in size. We follow its original paper to use $4$ message passing steps at the top and bottom levels and two for the others.

- **BSMS-GNN.**    We use the same number of graph hierarchies in EvoMesh and as in BSMS-GNN. We use the minimum average distance as the seeding heuristic for the BFS search recommended in its original paper. The multi-level building is processed in one pass. The inter-level propagation uses the normalized node degree to convolve features from neighbors to central nodes. The encoder, decoder, node update function, and edge update function are set up the same way as in our model. We perform one message passing step at each graph level.

- **Eagle.**    We use the *same-size KMeans* algorithm in the preprocessing step with a cluster size of 10. The encoder consists of 4 stacked GNN layers. The graph pooling module comprises a single-layer gated recurrent unit followed by a single-layer MLP. The hidden dimension is set to 128 across all datasets. The attention module includes 4 sequential attention blocks, each with a single attention head. A final layer normalization is applied after the last attention block. The decoder shares the same architecture as that used in EvoMesh.

- **HCMT.**    The hidden dimension and the number of attention heads in the HCMT block are set to $128$ and $4$, respectively. We use the same number of hierarchies as in EvoMesh. For the Cylinder and Airfoil datasets, due to the presence of hollow sections in the mesh, we do not apply Delaunay triangulation for remeshing. Instead, we use edge connections generated through bi-stride pooling.Like in EvoMesh, we use a single message passing step at each graph level.

Notably, the node encoder, decoder, node update function, and edge update function of MGN, Lino *et al.*, BSMS-GNN, HCMT and Eagle have the same network architecture as those in EvoMesh. To reduce the number of network parameters, we avoid separately encoding the edge offset $\mathbf{e}_{ij}$. Instead, we concatenate it with the node latents and use this combined input for the edge update function to compute $\hat{\mathbf{e}}_{\mathbf{ij}}$.

All models are trained using the Adam optimizer for $1M$ steps, with an exponential learning rate decay from $10^{-4}$ to $10^{-6}$ and a decay rate of $\gamma = 0.79$ from the first $500K$ steps. The batch size is set to 32.

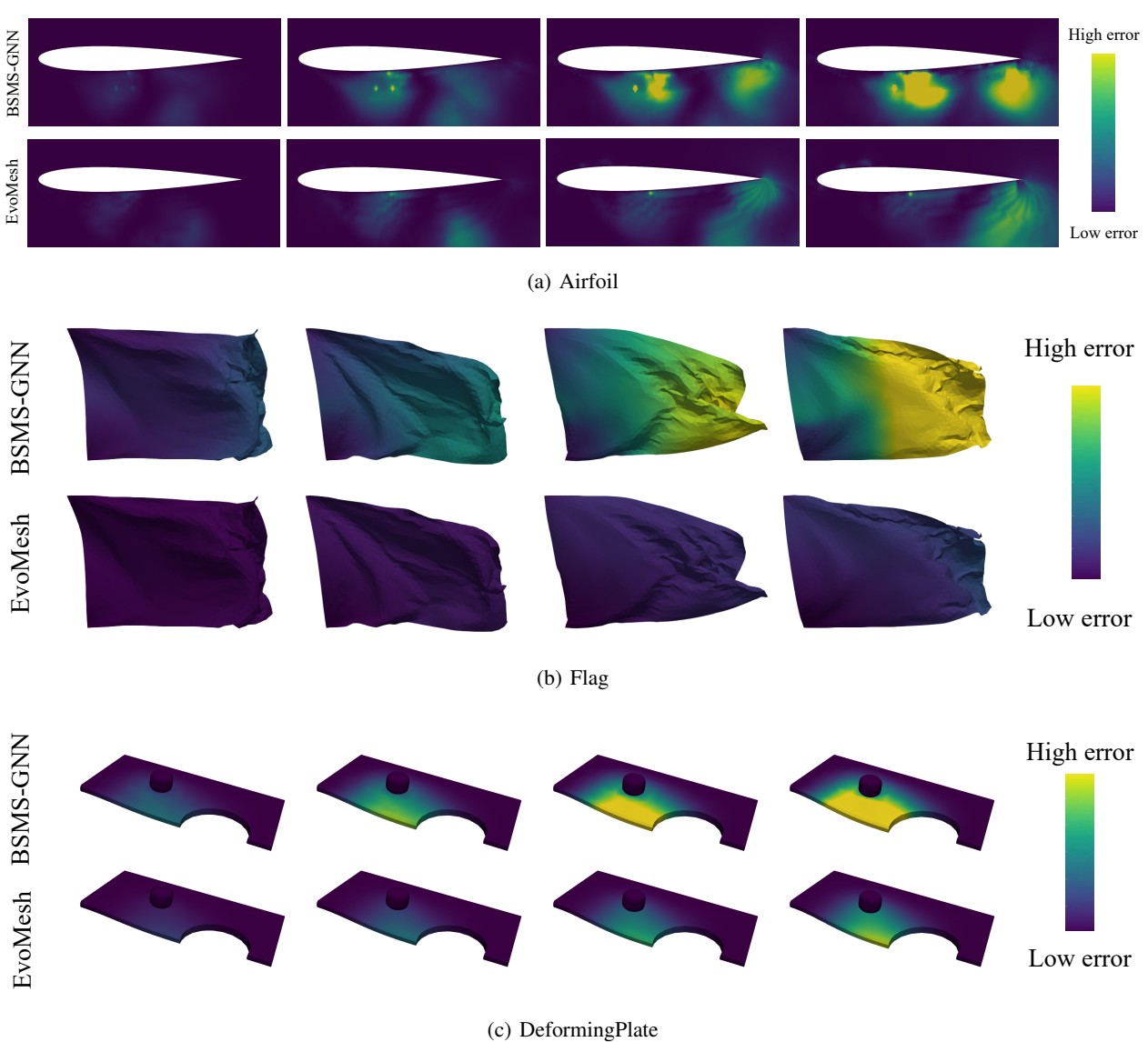

(a) Airfoil

(b) Flag

(c) DeformingPlate

*Figure 9.* Showcases of rollout prediction error maps on Airfoil, Flag and DeformingPlate dataset.

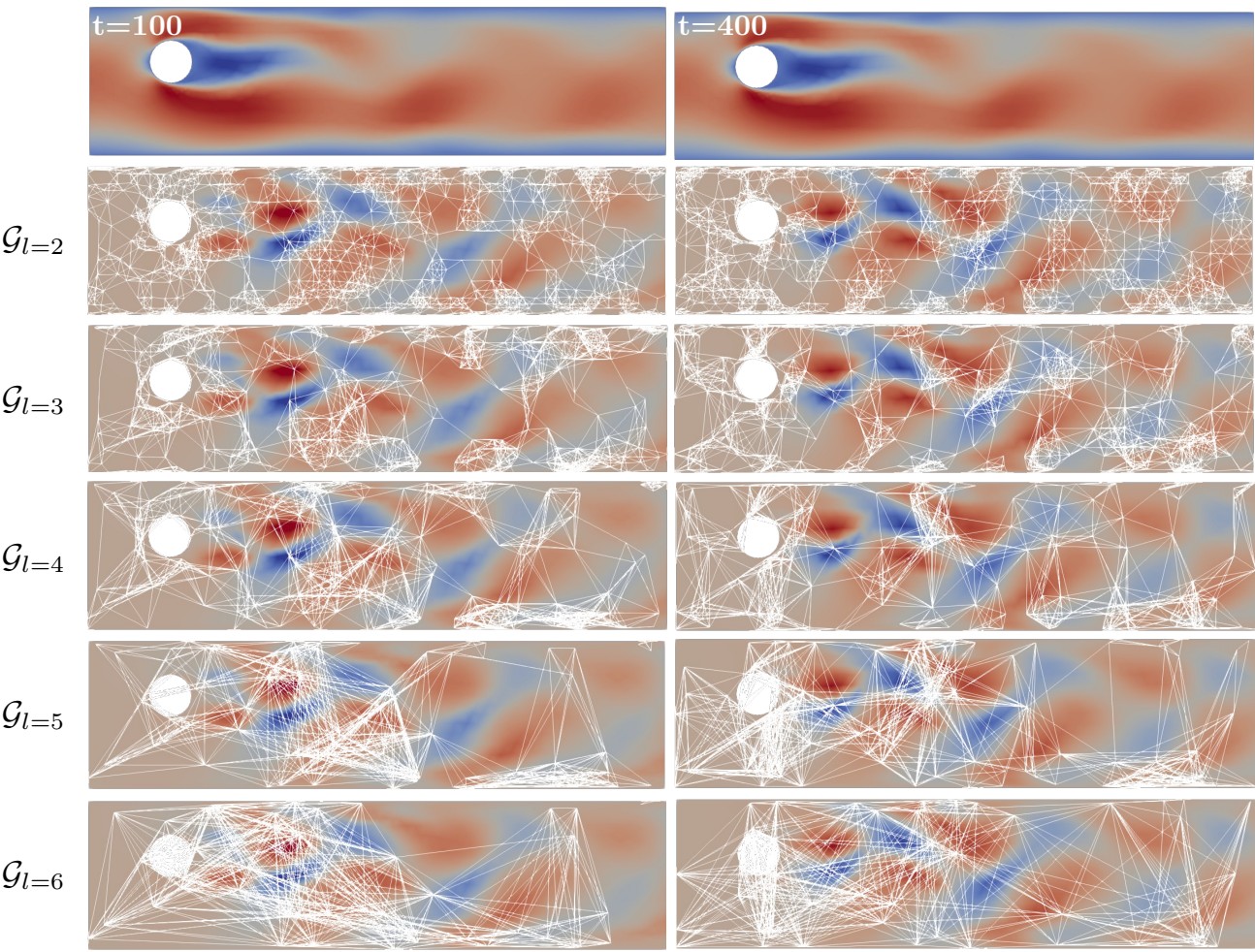

*Figure 10.* **Row 1:** The velocity field from the true data on the CylinderFlow dataset. **Row 2-6:** The temporal difference of the velocity fields between adjacent time steps alongside the constructed coarser-level graphs.

