# OpenReview forum: "EvoMesh: Adaptive Physical Simulation with Hierarchical Graph Evolutions"
_ICML.cc/2025/Conference — ICML 2025 poster_

### Official Review · Reviewer_oo3z · 2025-02-23

**Overall Recommendation:** 3

**Summary:**

This paper presents a hierarchical GNN method for learning PDEs. In contrast to many other such methods, it uses differentiable graph construction, so the message passing hierarchy is learned end-to-end.

**Claims And Evidence:**

Yes. The main claim is better accuracy compared to multi-scale GNNs with fixed hierarchy, and the paper does provide evidence for this.

**Essential References Not Discussed:**

GAT (see above).
There is also a bunch of literature on differentiable graph pooling (e.g. Ying et al's DiffPool), which should probably be discussed in related work.

**Experimental Designs Or Analyses:**

There's a few important comparisons & baselines missing.

1. I didn't see any test for the efficacy of AMP-- there should be an ablation of AMP against 'isotropic' summarization
2. It would be very nice to analyze/discuss number of levels/downsampling ratio for EvoMesh & the baselines. How different are the hierarchies used by the baselines or the 'static' ablation? Is spatial location important, i.e. how does EvoMesh compare against a hierarchy with same amount of nodes per level, but sampled uniformly across the domain? Etc.
3. There should be a comparison against a single-level GNN (probably MGN, since most of the datasets are taken from that paper). I took a glance at the RMSE number in EvoMesh and the MGN paper, and RMSE reported for MGN seem actually lower for CylinderFlow, Airfoil, Plate? (it's possible there are differences in how those are computed though).

**Methods And Evaluation Criteria:**

Benchmark datasets are fine for the task. Although I probably would have included a dataset with fast-acting/long-range physics, for which single-level GNNs perform poorly and hierarchies are necessary.

I'm not really convinced by the motivation for AMP. Oversmoothing is particularly a problem for long-range information retrieval tasks, and much less of an issues for learned PDE solvers. Those generally provide geometry information in edge features, plus by definition in PDEs information does spread locally. Summation aggregation is also actually much closer to the underlying computations in physics than attention: see force superposition in Lagrangian and differential kernel computation in the Eulerian view.

**Other Comments Or Suggestions:**

### After rebuttall ###

The additional experiments (especially vs MGN, GAT) strengthen the paper, and hence I have improved my score. A paper with all the additional experimental results and updated explanations might meet the bar for ICML.

But there's still a bunch of unanswered questions around differentiability: differentiable sampling with GS for graph construction seems to be rarely done, yet somehow this paper seems to make it work without any major new innovation. It was my understanding that with GS sampling gradient quality wouldn't be sufficient, and most papers in this space I'm aware of use soft sampling with all edges active.
Also I'm not sure the role of AMP is properly addressed-- in the rebuttal the authors pose the theory that while it might not necessarily help with physics it helps with node selection. It would be nice to properly investigate this (and the relation to GAT).

So while I wouldn't be opposed to having the paper published, I do feel that with taking a step back, a performing a major revision, this could be a much better paper, which provides much more value to the ML community.

###

**Other Strengths And Weaknesses:**

Strengths:
I like the premise of the paper-- there's clear evidence that for GNNs, multi-scale hierarchies are important for many problems in physics. But the optimal way to construct these hierarchies is still TBD, and being able to perform differentiable end-to-end graph construction would be very impactful.

Weaknesses:
1. As mentioned above, part of the contribution may not be novel.
2. In a way, node selection is the easy part. The paper needs to explain how it gets around the issue of differentiable edge graph construction -- fully connected graphs for gl>1 are very expensive, which is often cited as the reason that fully differentiable end-to-end graph construction is not viable.
3. The paper needs a bit more thorough evaluation (see missing experiments above)

**Questions For Authors:**

- From fig 10, it looks like the downsampling ratio is around ~2 per level. Is there anything in the model that encourages a particular ratio, or put pressure to use less points? What's stopping the model from retaining every single point?

- Differentiable node sampling makes sense; however, what about edge construction? Differentiable pooling often uses fully connected graphs, with down-weighting for inactive edges, to make this process differentiable. But it sounds like this paper doesn't use fully connected graphs for G_l>1-- how is differentiability guaranteed?

These points also need to be described more explicitly in the paper.

**Relation To Broader Scientific Literature:**

AMP looks _very_ similar, if not identical, to Graph Attention Networks (Velickovic et al). So either this should be removed as a contribution, or if the authors believe that it is sufficiently different, GAT needs to be discussed & compared to, and notable differences pointed out.

**Theoretical Claims:**

N/A

---

> ### Author Rebuttal · Authors · 2025-04-01
>
> > Q1: Differentiability for $G_{l>1}$.
>
> EvoMesh avoids using fully connected graphs for ${G}_{l>1}$. Instead, it ensures differentiability through:
> 1. Node selection: Differentiable via Gumbel-Softmax (L183-197).
> 2. Edge construction: Edges are formed based on the selected nodes and the connectivity of the original graph, further enhanced by $K$-hops (L199-219).
> 3. Edge feature generation: A differentiable process that encodes distance vectors and norms between connected nodes into high-dimensional edge features, serving as inputs for edge and node updates in message passing.
>
> As a result, the prediction error can be backpropagated to both the generated edges and selected node features, allowing the node selection module to be optimized.
>
> > Q2: Novelty beyond AMP.
>
> We appreciate the reviewer’s focus on evaluating the rationale behind AMP. Meanwhile, we'd like to emphasize that the primary novelty of EvoMesh lies in its **learnable dynamic graph structures**, with AMP being one component for enabling this capability. We hope the reviewer will consider the broader innovation of our work as a whole.
> * **Core Contribution - Learnable Dynamic Graph hierarchies**: Unlike traditional PDE solvers that rely on fixed/predefined graphs, EvoMesh introduces a differentiable method to learn time-evolving graph structures that adapts to physical system dynamics.
> * **Role of AMP**: AMP enables anisotropic intra-level propagation to determine node selection; and its predicted importance weights facilitate inter-level feature propagation.
>
> > Q3: Efficacy of AMP.
>
> The efficacy of AMP is demonstrated in two ways:
> - Fig 4 (Inter-level): By comparing the model with M1-M3 baselines, it suggests that AMP's predicted importance weights are crucial for inter-level information propagation.
> - Anisotropic vs. Isotropic (Intra-level): We here use isotropic summation for intra-level feature aggregation, while retaining the AMP importance weights for inter-level propagation. The results are shown below:
>
> ||RMSE-1($10^{-2}$)||RMSE-All($10^{-2}$)||
> |-|-|-|-|-|
> ||Cylinder|Flag|Cylinder|Flag|
> |Isotropic|0.4521|0.3752|40.59|118.9|
> |EvoMesh|0.1568|0.3049|6.571|76.16|
>
> > Q4: Number of levels/downsampling ratio.
>
> Number of hierarchy levels: We use the same number of layers as the static bi-stride hierarchy method. Please refer to **Appendix C.4** for the impact of varying hierarchy numbers.
>
> Downsampling ratio: Below, we present the average downsampling ratios for EvoMesh and the static bi-stride hierarchies. We visualize static and EvoMesh hierarchies at https://sites.google.com/view/evomesh/.
>
> |Node retention|Cylinder||Flag||
> |-|-|-|-|-|
> |Layer|Static|EvoMesh|Static|EvoMesh|
> | 1| 0.5029| 0.4510| 0.5009| 0.4409|
> | 2| 0.5069| 0.5090| 0.5082| 0.6288|
> | 3| 0.5147| 0.5294| 0.5174| 0.5251|
> | 4| 0.5356| 0.5031| 0.5654| 0.7323|
> | 5| 0.5131| 0.4306| 0.5582| 0.2626|
> | 6| 0.5039| 0.6331| 0.5129| 0.7182|
>
> > Q5: vs. Uniform node-sampling.
>
> We present the required comparisons below. The baseline model uses an equal number of uniformly sampled nodes at each level.
>
> ||RMSE-1($10^{-2}$)||RMSE-All($10^{-2}$)||
> |-|-|-|-|-|
> ||Cylinder|Flag|Cylinder|Flag|
> |Uniform sample|0.3019|0.3999|9.357|145.27|
> |EvoMesh|0.1568|0.3049|6.571|76.16|
>
> > Q6: MGN reports lower RMSE for CylinderFlow/Airfoil/Plate.
>
> Most models converge after 1 million steps, while MGN trains for 10 million steps (10 days for CylinderFlow, 20 for Airfoil). To fix this, we use 500 trajectories for the Plate dataset, whereas MGN paper uses 1000. For Airfoil, our sequence length is 100 steps, while MGN uses 600 (detailed in Appendix). We now include MGN results in our response to **Reviewer Mnuf Q4**, showing EvoMesh outperforms MGN.
>
> > Q7: AMP vs. GAT.
>
> Please see our response to **Reviewer Mnuf Q3**.
>
> > Q8: Related work -- DiffPool.
>
> We'll include a discussion of these methods in the revised paper -- Most differentiable graph pooling methods like DiffPool are designed for graph classification, where unpooling is not typically needed. In contrast, EvoMesh focuses on mesh-based physics simulation, requiring dynamic modeling across multiple scales and integration of global and local details. Dynamic hierarchy construction remains largely under-explored in this field.
>
> > Q9: Ratios of retained points.
>
> We provided sampling ratios of EvoMesh in our previous response to Q4, which shows variations across different graph layers. While static methods follow predefined reduction steps, EvoMesh adaptively retains nodes **based on the optimization objective**, which is solely aimed at improving prediction accuracy. We do not impose constraints on sampling ratios.
>
> Is it possible for the model to retain all nodes? Theoretically, yes. However, in practice, the model prioritizes the most relevant nodes for the simulation, those that contribute the most to the final prediction. This behavior emerges naturally during training, suggesting that dense high-level graph structures may not be necessary for this task.

---

### Official Review · Reviewer_2MCy · 2025-03-11

**Overall Recommendation:** 3

**Summary:**

This paper proposes anisotropic message passing (AMP) with hierarchical structure for mesh-based simulation. Specifically, the AMP enables	GNN to predict the edges’ weights before aggregation. The hierarchical graphs are constructed dynamically given the predicted importance. Experiments on five different domains show that the proposed method outperforms baselines.

## Update after rebuttal
The author has addressed my concerns, so I remain positive about this paper and will keep my original score.

**Claims And Evidence:**

The claims are clear.

**Essential References Not Discussed:**

The paper should discuss LayersNet [1], which also deals with learning-based simulation. The relation and similarity should be discussed. For example, LayersNet also applies hierarchical structure for mesh-based simulation; the attention mechanics and the AMP's selection share similarity.

[1] Shao, et al. Towards Multi-Layered 3D Garments Animation, ICCV2023.

**Experimental Designs Or Analyses:**

The experiments are sufficient. Visualized results on all domains are provided either in the main text or appendix. Both the mean and standard deviation are reported in the quantitative results.

**Methods And Evaluation Criteria:**

Make sense.

**Other Comments Or Suggestions:**

Please refer to "Questions For Authors".

**Other Strengths And Weaknesses:**

The experiments are sufficient and convincing. It would be better if the author could provide more visualized results, such as videos with more frames.

**Questions For Authors:**

Here I summarize and list my questions:

1. Is the proposed method bring extra computational overhead? What is the efficiency of this method?
2. Missing reference in "Essential References Not Discussed".

**Relation To Broader Scientific Literature:**

Please refer to "Essential References Not Discussed".

**Theoretical Claims:**

N/A. The claims are generally correct.

---

> ### Author Rebuttal · Authors · 2025-04-01
>
> We thank the reviewer for the insightful comments. Below, we address each comment point-by-point.
>
> > Q1: Missing reference -- LayersNet.
>
> Thank you for bringing LayersNet to our attention. We will include it in our related work and provide detailed comparisons in the revised manuscript. Below is a brief discussion of the key differences:
> - *Distinct application domains:* While LayersNet focuses on garment animation influenced by external forces, EvoMesh is a general mesh-based simulation method for both Eulerian (e.g., fluid and structure simulations) and Lagrangian (e.g., cloth simulation) systems.
> - *Different hierarchical structures:* LayersNet employs a **static, patch-based two-level hierarchy**, where garments are represented as particle patches to simplify interaction modeling. In contrast, EvoMesh introduces a **learning-based, time-evolving graph structure that dynamically adapts to physical system dynamics**, with the entire structure being learned in a fully differentiable manner.
> - *Attention vs. AMP mechanisms:* LayersNet employs specialized attention mechanisms to handle rotational symmetry in garment dynamics. In EvoMesh, **AMP serves as the core mechanism for dynamic graph construction**, where the differentiable node selection is guided by AMP features. Additionally, AMP’s predicted importance weights facilitate inter-level feature propagation. We also discuss the impact of replacing AMP with GATConv, which results in degraded performance, as detailed in our response to **Reviewer Mnuf Q3**.
>
> > Q2: More visualized results.
>
> We provide additional visualizations in https://sites.google.com/view/evomesh/, including prediction showcases of FoldingPaper, constructed comparisons for static-based bi-stride hierarchy and dynamics hierarchies of EvoMesh. We hope this addresses your concern.
>
> > Q3: Is the proposed method bring extra computational overhead?
>
> We have provided a comprehensive efficiency comparison in **Appendix Table 13**, covering training time, inference time, and model size.
>
> Admittedly, the dynamic hierarchy construction in EvoMesh does introduce some additional computational overhead. However, as shown in Table 13, this cost is manageable as **the overall training time remains comparable to that of static hierarchy methods**. This efficiency is primarily attributed to the significant reduction in the number of edges at each hierarchy level, as detailed in our response to **Reviewer TDvH Q7**.

---

### Official Review · Reviewer_TDvH · 2025-03-11

**Overall Recommendation:** 4

**Summary:**

The paper presents a novel hierarchical graph network architecture in which the hierarchy is determined in a data-driven manner. Additionally, it introduces an anisotropic message-passing step that incorporates an attention mechanism into the aggregation process. With these innovations, the proposed EvoMesh architecture outperforms various hierarchical baselines across multiple challenging tasks by a significant margin.

**Claims And Evidence:**

All claims are backed up by evidence.

**Essential References Not Discussed:**

Not the case.

**Experimental Designs Or Analyses:**

The selection of only three seeds seems somewhat low. However, the standard deviations suggest that significant variations between different training runs are unlikely.

**Methods And Evaluation Criteria:**

## Method
I did not fully understand the details of the DiffSELECT operation. The model needs to sample nodes during training, and through reparameterization, the gradient somehow propagates through this process. However, the exact mathematical formulation of Eq. (6) remains unclear from reading only the current paper. Providing a precise formulation—perhaps in the appendix—would improve the paper, as this is a crucial component enabling the data-driven hierarchical selection.

Similarly, the motivation behind FeatureMixing is not entirely clear. Is it intended to combine features from a coarser level with those from the current level? A more detailed explanation would help clarify this aspect of the architecture.

## Evaluation
The evaluation appears thorough, covering a range of ablations and baselines. However, I missed a complete rollout from the OOD mesh resolutions—or, if a full rollout is too noisy, a multi-step rollout could still be informative. Additionally, a qualitative showcase of the paper-folding task would be valuable. Since, to my understanding, this is a new task and dataset, visualizing the results would help illustrate the model's performance more effectively.

**Other Comments Or Suggestions:**

- In Figure 2, the authors speak from “DHMP”, I suspect that this is an older acronym of the EvoMesh method?
- Same goes for Figure 8a.

**Other Strengths And Weaknesses:**

The introduction effectively communicates the problem, the proposed method, and its broader context. Additionally, the analysis and evaluation are thorough. I enjoyed reading the paper. However, as mentioned earlier, some details in the method description—particularly regarding the DiffSELECT operation and the FeatureMixing method—are missing. Expanding on these aspects would improve clarity.

**Questions For Authors:**

- According to the appendix, the runtime was similar to that of static hierarchy models. Why is this the case? Given that hierarchy construction is typically costly, did the authors employ any optimizations or tricks to improve efficiency? Clarifying this would be helpful.
- Will the authors release their code and datasets? Given the complexity of the data-driven hierarchy implementation, making the code available would help increase adoption of the method. Additionally, I would appreciate the opportunity to compare it more easily in my own work. I consider raising my score to Strong Accept if that is the case.

**Relation To Broader Scientific Literature:**

This work builds upon hierarchical graph network simulators, introducing novel aspects in both anisotropic message passing and data-driven hierarchy construction. While Attentive Graph Neural Networks share some similarities with the anisotropic message passing steps, their structure differs significantly, making this a clear contribution.

**Theoretical Claims:**

No theoretical claims were made.

---

> ### Author Rebuttal · Authors · 2025-04-01
>
> We greatly appreciate the reviewer’s valuable comments.
>
> > Q1: Details of DiffSELECT.
>
> We here revise Eq. (6) to clarify the node selection process. Specifically, we employ Gumbel-Softmax independently for each node based on a 2-dimensional logits vector predicted by model $\phi^v$. This enables stochastic sampling of node retention in the downsampled graph while ensuring differentiability for gradient-based optimization.
>
> The formulation is as follows:
>
> $$
> z\_i^l=\text{Gumbel-Softmax}(\mathbf{l}\_i^l)=\frac{\exp\left((\log p\_{i,0}^l+g\_{i,0}^l) / \tau \right)}{\sum_{k=0}^1 \exp\left((\log p_{i,k}^l + g_{i,k}^l) / \tau \right)},
> $$
>
> where:
> - $\mathbf{l}\_i^l=(\log p\_{i,0}^l, \log p\_{i,1}^l)$ represents the logits for node $v_i$ at layer $l$,
> - $g_{i,k}^l$ is the Gumbel noise sampled independently for each node and category,
> - $\tau$ is the temperature parameter controlling the smoothness of the sampling process.
>
> This process allows for a **differentiable approximation of discrete node selection**. We will include this explanation in the revised manuscript.
> > Q2: On FeatureMixing.
>
> FeatureMixing is to integrate features from finer levels with those from coarser levels, to better capture both local and global information in the U-Net architecture. While the EXPAND operation upsamples coarser-level features (e.g., level $L$) to match the resolution of the current level ($L−1$), naive upsampling can introduce noise or lead to uneven upsampling. FeatureMixing refines and aligns coarser-level features before fusion.
> The whole upsampling process works as follows:
> 1. **EXPAND**: Coarser-level features ($\mathbf{F}_L$) are upsampled to the resolution of level $L-1$.
> 2. **Refinement**: A message-passing step is applied to refine the upsampled features ($\text{EXPAND}(\mathbf{F}_L)$):
>      $\mathbf{\tilde{F}}_L = \text{MessagePassing}(\text{EXPAND}(\mathbf{F}_L))$
> 3. **Fusion**: The refined coarser-level features ($\mathbf{\tilde{F}}\_L$) are combined with the current level’s intra-level features ($\mathbf{F}\_{L-1}$):
>    $\mathbf{F}\_{L-1}^{\text{mixed}} = \mathbf{F}_{L-1} + \mathbf{\tilde{F}}_L$
>
> We will improve the clarity of FeatureMixing in the revision.
> > Q3: Multi-step rollouts on OOD Mesh Task.
>
> We have supplemented the rollout predictions with 50 steps and results of additional baselines on the Cylinder and Airfoil datasets in the rebuttal period. Results are presented in the reply to **Reviewer Mnuf Q4**. As shown, EvoMesh outperforms other methods in both the 1-step and 50-step rollout predictions, demonstrating superior generalization.
> > Q4: Qualitative showcases of paper-folding.
>
> Visualizations of EvoMesh and MGN rollouts are available in https://sites.google.com/view/evomesh/. EvoMesh better captures fine details and deformation, adapting to intricate folds and complex geometries. We hope this addresses your concern.
> > Q5: Gaussian input noise.
>
> The values for Gaussian input noise are reported in Appendix A, Table 7. We adopted the same noise scale used in the MGN and BSMS-GNN papers for each dataset, without task-specific tuning. The MGN paper and its referenced GNS paper[1] offer a detailed discussion on the selection of the Gaussian noise scale, and subsequent work generally adhered to these established values. The same scale is applied in the OOD task, regardless of mesh resolution.
>
> [1] Learning to Simulate Complex Physics with Graph Networks.
> > Q6: Typo in Figure 2 and Figure 8a.
>
> "DHMP" indeed refers to EvoMesh, and we'll correct this in the revised manuscript.
>
> > Q7: Why similar runtime to static hierarchy models? Any optimizations to improve efficiency?
>
> EvoMesh maintains a runtime comparable to static hierarchy models because the computational overhead of hierarchy construction is *balanced by a substantial reduction in the number of edges* at each hierarchical level. As shown below, EvoMesh consistently yields fewer edges per layer compared to the static bi-stride hierarchy. Since message passing complexity is primarily driven by the number of edges rather than nodes, this reduction substantially lowers computational costs.
>
> Notably, EvoMesh achieves this without explicit efficiency-driven modifications. Instead, this efficiency emerges naturally during training, as EvoMesh autonomously identifies key nodes, which we observe to be relatively sparse. This partially suggests that the dense high-level graph structures may not be necessary for this task.
>
> || CylinderFlow (#nodes, #edges)||(#nodes, #edges)||
> |-|-|-|-|-|
> |Layer|Static |EvoMesh| Static|EvoMesh|
> |1|(949, 8128)|(851, 5322)|(791, 7130)|(696, 4347)|
> |2|(481, 6333)|(433, 3150)|(402, 5728)|(438, 3975)|
> |3|(247, 5308)|(229, 2021)|(208, 5210)|(230, 2681)|
> |4|(133, 4645)|(115, 1172)|(118, 5654)|(168, 2178)|
> |5|(81, 4285)|(50, 484)|(77, 4923)|(44, 588)|
> |6|(41, 1654)|(31, 358)|(40, 1537)|(29, 174)|
>
> > Q8: Code release.
>
> We'll release both the code and datasets upon acceptance.

---

### Official Review · Reviewer_Mnuf · 2025-03-14

**Overall Recommendation:** 2

**Summary:**

The paper presents EvoMesh, a graph neural network for mesh-based physical simulations that dynamically learns evolving graph hierarchies instead of relying on fixed structures. Using anisotropic message passing, it adaptively selects nodes based on physical inputs, improving long-range dependency modeling. Experiments show improvement over fixed-hierarchy GNNs across multiple simulation datasets.

**Claims And Evidence:**

Yes

**Essential References Not Discussed:**

The related work seems include sufficiency references.

**Experimental Designs Or Analyses:**

Yes, seems reasonable.

**Methods And Evaluation Criteria:**

Make sense.

**Other Comments Or Suggestions:**

Please find comments in the question section.

**Other Strengths And Weaknesses:**

Overall, the proposed approach appears reasonable, and the results show improvement over baseline methods. However, a primary concern is that the novelty of the method seems limited, as it closely resembles the BSMS-GNN in overall structure.

**Questions For Authors:**

1. Could the authors provide more detail about the differences or novel aspects of the proposed approach compared to BSMS-GNN?
2. BSMS-GNN’s main contribution is its computational efficiency, which may come at the expense of prediction accuracy. In contrast, methods like EAGLE (Janny et al.) emphasize multi-scale message-passing algorithms and typically achieve better predictive performance.
The authors might consider comparing their approach to such methods.
3. The anisotropic message-passing layer appears similar to a variant of GAT. Could the authors explain this resemblance in more detail?
4. There are inconsistencies in the choice of methods used for comparison in different experimental setups. For example, why is MGN not included in Table 2 but only in Table 3? Likewise, why are methods like MGN and Lino et al. (2022) excluded from Table 4 comparisons?

**Relation To Broader Scientific Literature:**

The paper advances GNN-based physical simulations by replacing fixed graph hierarchies with adaptive, time-evolving structures via differentiable node selection.

**Theoretical Claims:**

Yes, seems correct

---

> ### Author Rebuttal · Authors · 2025-04-01
>
> We thank the reviewer for the insightful comments.
>
> > Q1: Novelty compared to BSMS-GNN.
>
> While both EvoMesh and BSMS-GNN adopt U-Net-based hierarchical structures for multi-scale modeling, EvoMesh introduces critical innovations in **end-to-end joint learning of graph hierarchies and physics dynamics that adapt to changing physics states**:
>
> **Core Innovation: Adaptive vs. Static (preprocessed) Hierarchies**
> - *EvoMesh* integrates graph structure learning and dynamics learning into a fully differentiable framework, enabling **time-evolving graph hierarchies that adapt dynamically to changing physical conditions**. Specifically, this is implemented with *anisotropic intra-level propagation*, where edge weights are modulated by local physical states to enable directionally sensitive message passing, and *learnable inter-level propagation*, which adaptively learns interactions between hierarchical levels to optimize information flow across scales.
> - *BSMS-GNN* separates hierarchy construction (preprocessing) from dynamics learning. It employs a **fixed, pre-defined graph hierarchy** with static connectivity throughout the simulation, limiting its ability to refine its structure in temporally evolving systems.
>
> >Q2: EvoMesh vs. EAGLE (Janny et al.).
>
> In methodology,
> - *EAGLE* uses a *two-scale* hierarchical message-passing approach, downscaling mesh resolution via *geometric* clustering of mesh positions. **The fixed-size clustering is precomputed with a modified k-means algorithm, independent of dynamics modeling.**
> - *EvoMesh* constructs a *multi-scale* hierarchy that enables the **joint learning of adapative high-level graph structures and physical dynamics** within an end-to-end differentiable framework.
>
> The table below compares the performance of EAGLE and EvoMesh. The results show that EAGLE’s purely geometric two-scale hierarchy results in slightly lower predictive performance, while EvoMesh benefits from its context-aware dynamic hierarchy.
>
> | | RMSE-1 ($\times 10^{-2}$)| | RMSE-All ($\times 10^{-2}$) | |
> |-|-|-|-|-|
> || Cylinder| Flag | Cylinder| Flag |
> |EAGLE| 0.1733|0.3805| 20.05|127.7|
> | EvoMesh | 0.1568| 0.3049 | 6.571| 76.16 |
>
> > Q3: AMP vs. GAT
>
> The key distinction between AMP and GAT is that **AMP enables differentiable dynamic hierarchy construction**, a capability not supported by GAT. While AMP shares similarities with attention-based methods like GAT in computing importance weights, the detailed implementation differences are as follows:
> - GAT applies computed weights to aggregate node features through weighted summation;
> - Ours: (a) directly applies predicted weights to edge features, (b) leverages weights for inter-level feature propagation.
>
> This dual mechanism enables AMP to dynamically learn graph hierarchies adaptable to evolving physical systems while jointly modeling physics dynamics through gradient-based learning, rather than relying on static hierarchical structures.
>
> To further demonstrate the effectiveness of AMP, we replaced it with GATConv in EvoMesh, using a single attention head. **The results below show that AMP layers used in EvoMesh outperform GATConv.** Full comparisons between AMP and GAT will be included in the revised manuscript.
>
> | | RMSE-1 ($\times 10^{-2}$)| |RMSE-All ($\times 10^{-2}$) ||
> |-|-|-|-|-|
> ||Cylinder| Flag |Cylinder| Flag|
> | EvoMesh-GATConv|0.2025|0.3009|10.253| 106.2|
> | EvoMesh |0.1568|0.3049|6.571|76.16|
>
> > Q4: There are inconsistencies in the choice of methods used for comparison in different experimental setups.
>
> **Table 2:** Here, we present the results of MGN, showing that EvoMesh achieves superior performance.
>
> || RMSE-1 ($\times 10^{-2}$)||||RMSE-All ($\times 10^{-2}$) ||||
> |-|-|-|-|-|-|:-:|:-:|:-:|
> || Cylinder | Airfoil |  Flag  | Plate| Cylinder |Airfoil|Flag|Plate|
> |MGN|  0.3046|77.38| 0.3459 | 0.0579 |59.78| 2816|115.3|3.982|
> |EvoMesh|0.1568| 41.41| 0.3049 | 0.0282 |6.571 |2002|76.16| 1.296 |
>
> **Table 3:** For the Paper Folding dataset, fixed hierarchy-based approaches (e.g., BSMS-GNN and HCMT) are excluded because their hierarchies, derived from the initial graph input, cannot be applied across the entire sequence as the mesh structure evolves.
>
> **Table 4:** We have supplemented this table with additional baseline models in the rebuttal period. Additionally, we present results from 50-step rollouts to offer further insights into the performance of extended simulations. The results demonstrate the generalizability of EvoMesh to OOD mesh structures through dynamic hierarchy construction. Full results will be provided in the revised paper.
>
> ||RMSE-1 ($\times 10^{-2}$)|| RMSE-50 ($\times 10^{-2}$) | |
> |-|-|-|-|-|
> | |Cylinder|Airfoil|Cylinder|Airfoil|
> | MGN (**New**) | 1.0596|169.577|7.8332|1829.1|
> | Lino et al. (**New**) |25.8930|144.35|65.207|1299.4|
> | BSMS-GNN|  0.9177  |202.30 |2.0971|1677.3|
> | EvoMesh |  **0.4855**  | **126.70**|**1.0771**|**812.47**|

---

### Decision · Program_Chairs · 2025-05-01

**Decision:**

Accept (poster)

**Comment:**

The paper introduces a GNN model for mesh-based physical simulations based on dynamic graph hierarchies. The paper has merits that have been highlighted by the reviewers. These include the formulation of the problem and the good performance on standard benchmarks. However, the reviewers have also indicated limitations mainly related to the following points: (i) novelty of the methodology and relation to previous approaches (the AMP component); (ii) efficiency of the proposed methodology. Considering the author's responses, I am in favor of accepting the paper. I encourage the authors to take these comments into consideration and revise the manuscript accordingly to better position their work in relation to previously proposed approaches.